# Connexin 43 hemichannels regulate mitochondrial ATP generation, mobilization, and mitochondrial homeostasis against oxidative stress

Jingruo Zhang, Manuel A Riquelme, Rui Hua, Francisca M Acosta, Sumin Gu, Jean X Jiang*

Department of Biochemistry and Structural Biology, University of Texas Health Science Center, San Antonio, United States

**Abstract** Oxidative stress is a major risk factor that causes osteocyte cell death and bone loss. Prior studies primarily focus on the function of cell surface expressed Cx43 channels. Here, we reported a new role of mitochondrial Cx43 (mtCx43) and hemichannels (HCs) in modulating mitochondria homeostasis and function in bone osteocytes under oxidative stress. In murine long bone osteocyte-Y4 cells, the translocation of Cx43 to mitochondria was increased under $H_2O_2$-induced oxidative stress. $H_2O_2$ increased the mtCx43 level accompanied by elevated mtCx43 HC activity, determined by dye uptake assay. Cx43 knockdown (KD) by the CRISPR-Cas9 lentivirus system resulted in impairment of mitochondrial function, primarily manifested as decreased ATP production. Cx43 KD had reduced intracellular reactive oxidative species levels and mitochondrial membrane potential. Additionally, live-cell imaging results demonstrated that the proton flux was dependent on mtCx43 HCs because its activity was specifically inhibited by an antibody targeting Cx43 C-terminus. The co-localization and interaction of mtCx43 and ATP synthase subunit F (ATP5J2) were confirmed by Förster resonance energy transfer and a protein pull-down assay. Together, our study suggests that mtCx43 HCs regulate mitochondrial ATP generation by mediating $K^+$, $H^+$, and ATP transfer across the mitochondrial inner membrane and the interaction with mitochondrial ATP synthase, contributing to the maintenance of mitochondrial redox levels in response to oxidative stress.

*For correspondence: jiangj@uthscsa.edu

## Editor's evaluation

This fundamental work advances our understanding of the functional role of connexin43 in mitochondrial. The evidence support that it forms hemichannels at the mitochondrial inner membrane, and serves for optimal mitochondrial metabolism, which is enhanced by oxidant stress. The authors expanded their observation to another cell type and provided solid method explanations and experimental evidence on the quality of their mitochondria preparation.

## Introduction

The connexins (Cxs) family comprises 20 members in mice and 21 in humans (*Bedner et al., 2012*; *Söhl and Willecke, 2004*). Six Cx subunits form a hexamer as a hemichannel (HC), allowing molecules less than 1 kDa to pass through (*Goodenough and Paul, 2003*; *Jiang et al., 2007*). HCs function as a gate for molecular communication between intracellular and extracellular spaces, allowing the passage of molecules and the exchange of information across the membrane. Two HCs in adjacent cells can form a gap junction, which assists in cell-cell communication directly from the cytoplasm of adjoining cells,

thus synchronizing cellular functions (*Iyyathurai et al., 2013*). Under normal physiological conditions, undocked Cx HCs remain mainly closed (*Sáez et al., 2010*). However, pathophysiological conditions, such as injury or disease, drive their opening, leading to the augmentation of certain pathological conditions, such as increased inflammatory reactions (*Peng et al., 2022*). HC opening is stimulated by many factors like low extracellular $Ca^{2+}$ concentration, oxidative stress, mechanical signals, and extracellular alkalinization (*Burra and Jiang, 2011*; *Decrock et al., 2011*; *Francis et al., 1999*; *Gault et al., 2014*; *Kar et al., 2012*; *Leithe et al., 2018*; *Schalper et al., 2010*). The opening of Cx43 HCs also exhibits a positive impact on cells, especially in bone osteocytes (*Kar et al., 2013*; *Zhao et al., 2022*).

Cx 43 (Cx43) is the most ubiquitously expressed Cx in various cell types such as endothelial cells, cardiomyocytes, astrocytes, etc. It is also expressed in cellular organelles like mitochondria (*Fiori et al., 2014*; *Gu et al., 2006*; *Li et al., 2002*). Mitochondrial Cx43 (mtCx43) is orientated with the C-terminus facing the mitochondrial intermembrane space (*Miro-Casas et al., 2009*). The physiological role of Cx43 in mitochondria remains largely elusive. Some data suggest that mtCx43 regulates $K^+$ influx to the mitochondrial matrix, increasing mitochondrial respiration, ATP production, and reactive oxygen species (ROS) generation in cardiomyocytes (*Boengler et al., 2012*; *Miro-Casas et al., 2009*). The protective role mtCx43 plays in pre-conditioning may be benefited from the channel property of Cx43 HCs. Many ion channels located in mitochondria are involved in ensuring the ion exchange between mitochondrial membranes. Ions, including $Ca^{2+}$, $Na^+$, $K^+$, $H^+$, $Cl^-$, etc., not only participate in preserving normal mitochondrial membrane potential ($\Delta\Psi m$) but are also involved in regulatory pathways (*O'Rourke, 2007*). Thus, the ion-permeable channels are very important to normal mitochondrial function, especially ATP production. In mitochondria, the permeability of mtCx43 HCs for ions and molecules is dedicated to a mild uncoupling of the mitochondrial inner membrane, which eventually causes protection for the ischemia/reperfusion (I/R) process (*Ruiz-Meana et al., 2008*).

Osteocytes are the most abundant cell in bone, occupying 90–95% of adult total bone cells (*Bonewald, 2011*). Cx43 HCs, richly present in osteocytes, prominently mediate the anabolic action of mechanical loading and regulate the response to mechanical loading and oxidative stress (*Cherian et al., 2003*; *Hua et al., 2021*; *Zhao et al., 2022*). Oxidative stress causes apoptosis, further affecting the normal function of the osteocytic network. Cx43 HC is reported to reduce cell death induced by oxidative stress in osteocytic murine long bone osteocyte-Y4 (MLO-Y4) cells (*Kar et al., 2013*). Plasma membrane Cx43 HCs open under oxidative stress, which protects osteocytes from apoptosis. Oxidative stress induces the accumulation of membrane Cx43 HCs at the cell surface, alleviating the oxidative stress caused by the oxidant stimulation (*Kar et al., 2013*). The majority of prior studies focused on Cx43 HCs on the plasma membrane, while mtCx43 function in osteocytes under oxidative stress remains largely unexplored. This study provides new mechanistic insight into the role of mtCx43 and HCs in regulating mitochondria function and protecting cells against oxidative damage.

## Results

### Cx43 translocates to mitochondria and forms functional HCs

To elucidate the role of Cx43 in osteocytes under oxidative stress, we treated osteocyte MLO-Y4 cells with $H_2O_2$ and investigated the cellular distribution of Cx43. We found that the mitochondrial localization of Cx43 was increased by its fluorescence signal overlap with MitoTracker, a mitochondrial specific dye (*Figure 1A*). Manders' overlap coefficient analysis indicated a significant increase in co-localization between Cx43 and MitoTracker with $H_2O_2$ (*Figure 1B*). For further confirmation, succinate dehydrogenase (SDHA), a component of mitochondrial complex II, was also used as a marker for mitochondria as well. High-magnification images of isolated mitochondria further confirmed a clear overlap of SDHA and Cx43 fluorescence signals (yellow area; *Figure 1C*). Western blot analysis further showed the accumulation of Cx43 protein in mitochondria (*Figure 1D*), with a significant increase after 2 hr of treatment, peaking after 3 hr (*Figure 1E*). Purify of mitochondria preparation was examined using different organelle markers (*Figure 1—figure supplement 1*). To assess if the mtCx43 formed functional HCs, a dye-uptake assay was performed. Lucifer yellow (LY) dye (425 Da), which can permeate Cx43 HCs, was used as an indicator of active HCs, while tetramethylrhodamine (TRITC)-dextran (10 kDa), which is too large to pass through HCs, was used as a control to exclude non-specific dye uptake. The HC function was indicated by the ratio of LY/TRITC-dextran. Under standard conditions (Veh), the dye uptake in isolated mitochondria, following $H_2O_2$, was elevated

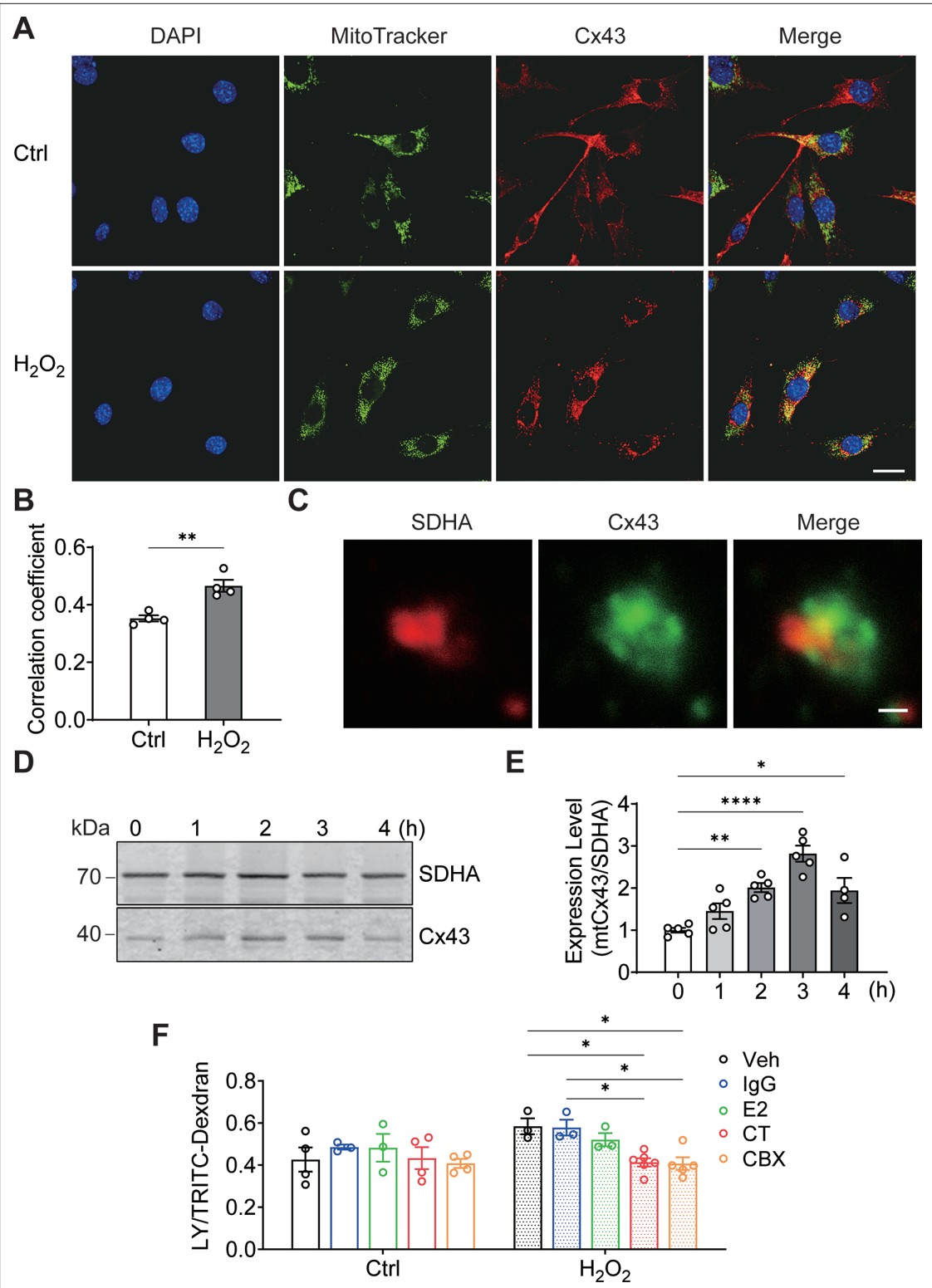

**Figure 1.** Connexin 43 (Cx43) translocated to mitochondrial and formed functional hemichannels (HCs) induced by oxidative stress in osteocytes. (**A**) $H_2O_2$ treatment induced the translocation of Cx43 to mitochondria in the murine long bone osteocyte-Y4 (MLO-Y4) cells. MLO-Y4 cells were treated with 0.3 mM $H_2O_2$ for 2 hr. After being stained with MitoTracker, cells were fixed and immunostained with Cx43(E2) antibody. Scale bar: 20 μm. (**B**) Manders' overlap coefficient co-localization analysis of MitoTracker and Cx43 based on fluorescence signals. Data collected from four independent experiments. Two-tailed t-test, **p<0.01. (**C**) Co-localization of succinate dehydrogenase (SDHA) and Cx43 in isolated mitochondria. Mitochondria were isolated from MLO-Y4 cells, fixed and subsequently immunostained with anti-Cx43 C-terminal (CT) and anti-SDHA antibodies. Scale bar: 1 μm. (**D**) Western

*Figure 1 continued on next page*

Figure 1 continued

blot and statistical data showed a significant increase of Cx43 in isolated mitochondria after 0.3 mM $H_2O_2$ treatment. One-way ANOVA analysis was conducted, **p<0.01. Data collected from N≥4 individual experiments. (**E**) Mitochondria dye uptake increased after $H_2O_2$ and were inhibited by Cx43 CT antibody and carbenoxolone (CBX). MLO-Y4 cells were treated with 0.3 mM $H_2O_2$ for 2 hr for mitochondria isolation. Mitochondrial dye uptake assay was performed with Lucifer yellow (LY)/TRITC-dextran in the absence and presence of 0.5 µg/mL Cx43CT, Cx43(E2) antibody or 1 µM CBX. N≥3, two-way ANOVA analysis was conducted, *p<0.05. TRITC-dexdran was used for calibration.

The online version of this article includes the following source data and figure supplement(s) for figure 1:

**Source data 1.** Connexin 43 (Cx43) translocation to mitochondria, raw data of co-localization analysis (**Figure 1B**), and western blot (WB) quantification (**Figure 1E**).

**Source data 2.** Original files of the WB images for **Figure 1D**.

**Source data 3.** Raw data of mitochondrial dye uptake for **Figure 1F**.

**Figure supplement 1.** The purity of mitochondria fractions isolated from murine long bone osteocyte-Y4 (MLO-Y4) cells.

**Figure supplement 1—source data 1.** Raw images for WB of mitochondria component purity in **Figure 1—figure supplement 1A**.

**Figure supplement 1—source data 2.** Raw data of the fluorescent signal counts in **Figure 1—figure supplement 1C**.

**Figure supplement 2.** Connexin 43 (Cx43) translocated to mitochondria in MC3T3-E1 osteoblast cell line after $H_2O_2$ treatment.

**Figure supplement 2—source data 1.** Raw data of co-localization analysis of MitoTracker and connexin 43 (Cx43) in MC3T3-E1 cell line.

(**Figure 1F**). To further confirm that functional mtCx43 HCs caused the increase in dye uptake induced by $H_2O_2$, carbenoxolone (CBX), a conventional Cx channel blocker, and Cx43 C-terminal (CT) antibody, which specifically binds to Cx43 C-terminus, were used. We used Cx43(CT) antibody to assess mtCx43 HCs because the C-terminus of mtCx43 faces the mitochondrial intermembrane space (**Boengler et al., 2009**; **Miro-Casas et al., 2009**). Cx43(E2) antibody, a specific Cx43 HC inhibitor targeting the extracellular loop of Cx43, and rabbit IgG, were used as controls (**Siller-Jackson et al., 2008**). We found that the LY uptake increment was inhibited by CBX and the Cx43(CT) antibody in mitochondria isolated from $H_2O_2$-treated cells, while Cx43(E2) and IgG antibodies had no effect on mtCx43 HCs (**Figure 1F**). Further confirmation was done with another cell line, osteoblastic cell, MC3T3-E1. We observed the similar translocation phenotype in response to $H_2O_2$ (**Figure 1—figure supplement 2**). These results suggest that Cx43 migrates to mitochondria, and mtCx43 HCs open in response to oxidative stress.

## Cx43 deficiency decreases mitochondrial membrane potential and ROS generation

By using the CRISPR-Cas9 lentivirus system, we KD *Gja1* in MLO-Y4 cells with *Rosa26* KD as a control. Whole-cell membrane and mitochondrial membrane proteins were extracted to evaluate the knock-down efficiency. An approximate 75% reduction in Cx43 protein level was seen from the total cell membrane (**Figure 2A**), and mitochondrial membrane protein (**Figure 2B**) extracts compared to the control group. Immunostaining of the isolated mitochondria from control and Cx43KD MLO-Y4 cells showed an overlap of SDHA and Cx43 proteins (orange signals), and the mtCx43 signal was much less in the Cx43KD group (**Figure 2C**). Meanwhile, the significant increase in dye uptake induced by $H_2O_2$ in control mitochondria was not seen in Cx43KD mitochondria (**Figure 2D**).

Mitochondrial electric membrane potential (ΔΨm), the main driver of oxidative phosphorylation (OXPHOS), is a key indicator of normal mitochondrial function. Tetramethyl rhodamine and ethyl ester (TMRE) was used to measure ΔΨm, and its depolarization indicated a damaged mitochondrial inner membrane stabilization (**Wang and Youle, 2009**). The TMRE intensity was significantly reduced in Cx43KD cells, indicating the depolarization and impairment of ΔΨm (**Figure 2E–F**). To mitigate the possible effects of the cytosol, we used mitochondrial complex specific inhibitors antimycin A (AA) and rotenone (Rot) to stimulate the generation of mitochondrial ROS (mtROS). We found that the basal level of mtROS was already decreased after Cx43 KD compared with its parallel control. AA and Rot were capable of inducing ROS generation in both control and Cx43KD mitochondria. However, the Cx43KD group generated less mitochondrial superoxide with the presence of inhibitors in intact cells, detected by mitoSOX (**Figure 2G**), and less ROS in isolated mitochondria, detected by DCF staining (**Figure 2H**). Together, the data suggest that Cx43 KD osteocytes had attenuated mitochondrial inner membrane potential and lower ROS generation.

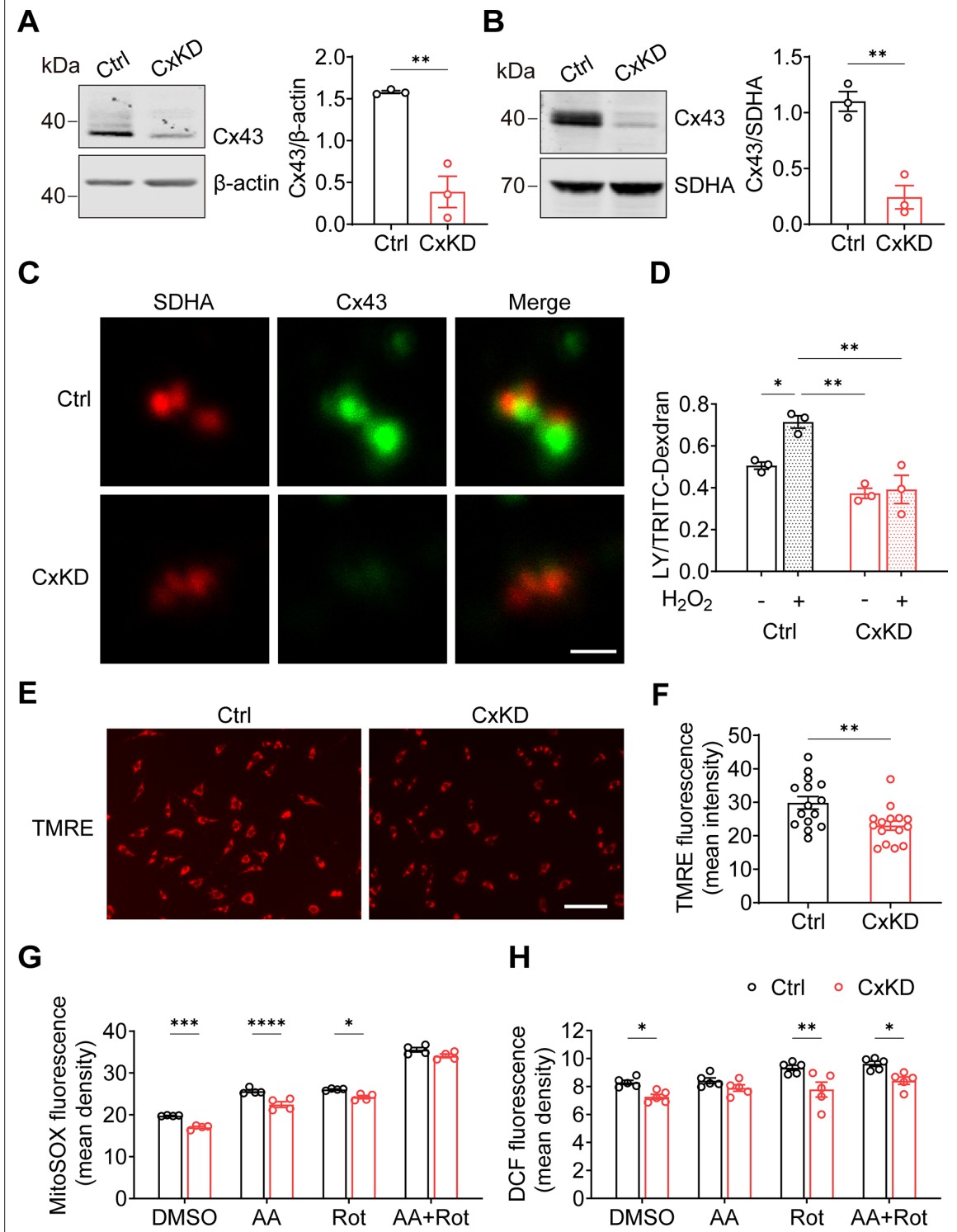

**Figure 2.** Connexin 43 (Cx43) knockdown (KD) reduced hemichannel (HC) function in mitochondria. (**A**) Cx43 was significantly knocked down in murine long bone osteocyte-Y4 (MLO-Y4) cells by the CRISPR-Cas9 system. Control and Cx43 KD (CxKD) MLO-Y4 cells were collected, and crude membrane extracts were analyzed using an anti-Cx43 antibody. Band intensity was quantified and compared by a two-tailed t-test (right panel). N=3; **p<0.01. (**B**) Cx43 was significantly knocked down in isolated mitochondria. Mitochondria were isolated from control (Ctrl) and CxKD MLO-Y4 cells and mitochondrial lysates immunoblotted with anti-Cx43 antibody. The KD efficiency in isolated mitochondria was quantified by the band intensity and compared by two-tailed t-test. N=3; *p<0.05 and **p<0.01. (**C**) Cx43 signal was absent in the mitochondria of Cx43KD MLO-Y4 cells. MLO-Y4 cells with or without Cx43 knockdown were immunolabeled with anti-Cx43 antibody and SDHA. Scale bar: 1 μm. (**D**) Dye uptake in isolated mitochondrial Cx43 KD MLO-Y4

*Figure 2 continued on next page*

*Figure 2 continued*

cells was abolished after H$_2$O$_2$ treatment. Mitochondria were isolated from MLO-Y4 cells with 2 hr 0.3 mM H$_2$O$_2$ incubation, and Lucifer yellow (LY)/RD dye uptake assay was conducted and quantified. (**E**) Mitochondrial membrane potential ($\Delta \phi$ m) was significantly decreased in the Cx43 KD MLO-Y4 cells. MLO-Y4 cells with or without Cx43 KD were incubated with tetramethyl rhodamine and ethyl ester (TMRE) which is used as an indicator for $\Delta \Psi$ m. TMRE fluorescence was shown in red. (**F**) TMRE fluorescence signals were detected and quantified (right panel). N=5 of independent experiments. Two-tailed t-test was used for statistical analysis. Each experiment was done in triplicates with three images from each repeat. ** p<0.01. Scale bar: 100 µm. (**G**) Cx43 KD significantly attenuated reactive oxygen species (ROS) production. MLO-Y4 cells with or without Cx43 KD were stained with MitoSOX, and MitoSOX fluorescence signals were quantified by NIH Image J and compared using two-way ANOVA, N=4, * p<0.05, and ** p<0.001. The cells were treated in the presence of complex inhibitors for 1 hr. AA: antimycin A and Rot: rotenone. (**G and H**) Mitochondrial ROS stained with fluorescent 2',7'-dichlorofluorescein (DCF) decreased in Cx43 impaired mitochondria. Mitochondria were isolated from either MLO-Y4 control or Cx43KD cells and treated with the inhibitors for 20 min. The fluorescence signals were measured using a microplate reader and quantified by NIH Image J. Two-way ANOVA analysis and multiple comparisons were conducted in each group, N=5. * p<0.05, ** p<0.001.

The online version of this article includes the following source data for figure 2:

**Source data 1.** Raw images for WB in *Figure 2A&B*.

**Source data 2.** WB quantification for *Figure 2A&B*.

**Source data 3.** Raw data of mitochondrial membrane potential for *Figure 2F*.

## Mitochondrial electron transport chain coupling is impaired after Cx43 KD

Mitochondrial electron transport chain (ETC) coupling with OXPHOS is a fundamental mechanism for ATP generation by utilizing oxygen (*Nolfi-Donegan et al., 2020*). The decreased $\Delta \Psi$ m is likely to impede mitochondrial OXPHOS. We performed an ETC coupling study. With the application of plasma membrane permeabilizer (PMP) in the assay medium, we conducted measurements in sequential order: basal respiration (state 2), after the injections of ADP (state 3), oligomycin (OLIGO) (state 4o), p-trifluoromethoxy carbonyl cyanide phenylhydrazine (FCCP) uncoupler (state 3u), and AA (*Figure 3A–B*). Oxygen consumption rate (OCR) in state 3 induced by ADP was significantly lower in Cx43KD cells compared with the control group. The maximal respiration rate by FCCP uncoupling in state 3u was also lower in Cx43KD cells compared to control. The non-mitochondrial OCR measured after injection of AA, a complex III inhibitor, was very low, almost to the same level as the control group. Both basal ATP production and ADP-induced ATP production were reduced in Cx43KD cells. Proton leak, measured by subtracting OCR after AA injection from state 4o, also showed a significant decrease in the Cx43KD group (*Figure 3C*). To demonstrate the ATP production level more directly, we extracted mitochondria and detected the ATP amount in the entire cell, mitochondria, and cytosol. A bioluminescence assay based on ATP recombinant luciferase and the substrate luciferin were used for ATP detection. ATP concentration was much less in the whole-cell, mitochondria, and cytosol of Cx43KD compared with control cells (*Figure 3D*). The consistent data generated from mitochondrial coupling and ATP detection assays indicated that ATP production in Cx43KD MLO-Y4 cells was decreased remarkably. In addition, quantitative PCR (qPCR) was conducted to further confirm if complexes in the mitochondrial respiration chain were affected at the mRNA level. We chose the marker proteins SDHA for complex II, ATP synthase subunit f (coded by *Atp5j2*) for complex V, and ADP/ATP translocase 2 (coded by *Slc25a5*) as its regulation of mitochondrial permeability transition pore (mPTP). The result showed the mRNA level of *Sdha* and *Atp5j2* was decreased in Cx43KD cells compared to the control group, while that of *Slc5a5* did not show a significant difference (*Figure 3E*).

## Cx43 HCs increase the proton gradient across the mitochondrial inner membrane

mtCx43 is reported to modulate K$^+$ influx into the mitochondrial matrix, and increased K$^+$ in the matrix promotes the respiration and efflux of protons from the matrix to intermembrane space (*Boengler et al., 2012*; *Heinen et al., 2007*). The proton gradient between the intermembrane space and matrix is a key factor for ATP generation. To determine the roles of mtCx43 HCs in the proton gradient under oxidative stress, we used pHluorin, a green fluorescent protein (GFP)-based fluorescent protein that is sensitive to H$^+$ signals, and live-cell imaging to detect proton changes. The process works by pH-insensitive blue fluorescent protein mTagBFP being fused before the N-terminus of pHluorin as a calibration, and the signal ratio of GFP/BFP increasing during the increment of pH (*Wu et al., 2019*).

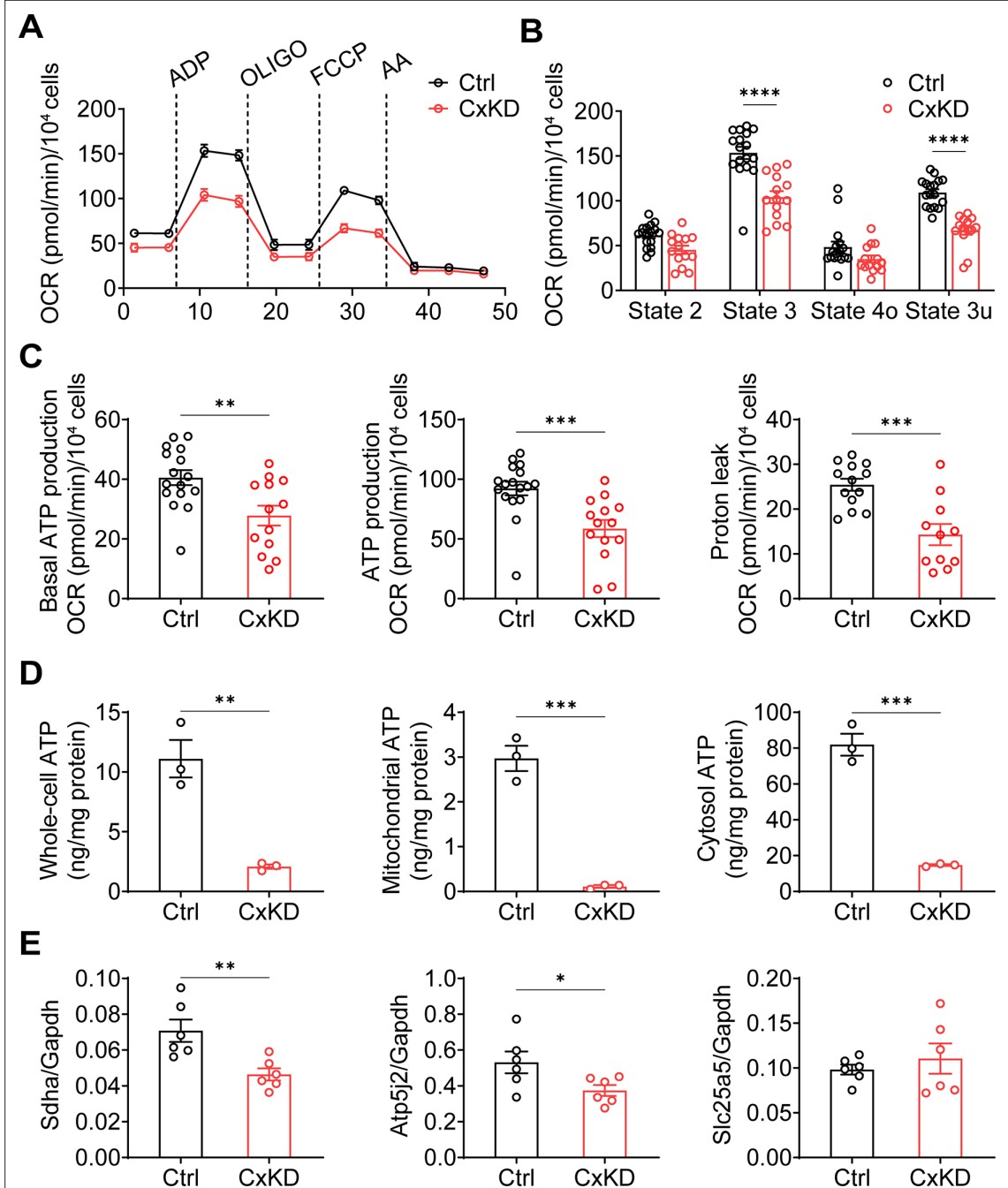

**Figure 3.** Mitochondrial functions were changed in the connexin 43 (Cx43) knockdown (KD) Y4 cell line. (**A**) The mitochondrial oxygen consumption was impaired in Cx43 KD Y4 cells. Seahorse XF Cell Mito Stress Test was used to determine mitochondrial function in cells. Oxygen consumption rate (OCR) was measured in each state in which ADP, oligomycin (OLIGO), p-trifluoromethoxy carbonyl cyanide phenylhydrazine (FCCP) and antimycin A (AA) were used to treat cells. OCR level calibrated based on 10,000 cells. (**B**) The OCR was significantly reduced after ADP and FCCP injections in Cx43KD Y4 cells. ***$p < 0.001$. (**C**) The proton leak and ADP-induced ATP production, calculated by OCR, was decreased in Cx43KD experimental groups. N≥3 repeats in each experiment, Two-tailed t-test, **$p < 0.01$, and ***$p < 0.001$. (**D**) ATP was dramatically decreased in Cx43KD Y4 cells. ATP determination was performed in isolated whole-cell lysis, isolated mitochondria, and cytosolic components. N=3. Two-tailed t-test, **$p < 0.01$, and ***$p < 0.001$. (**E**) The mRNA expression level of related genes in Y4 cell line. Mitochondrial complex II component protein succinate dehydrogenase (*Sdha*) and F(1)F(0) ATP synthase associated ATP synthase membrane subunit f (*Atp5j2*) were decreased in Cx43KD group. Mitochondrial permeable pore component ADP/ATP translocase 2 (*Slc25a5*) expression level had no significant changes between control and Cx43KD Y4 cells. N=6, Two-tailed t-test, *$p < 0.05$, and **$p < 0.01$.

The online version of this article includes the following source data for figure 3:

**Source data 1.** Raw data of oxygen consumption rate (OCR) in *Figure 3A–C*.

*Figure 3 continued on next page*

*Figure 3 continued*

**Source data 2.** Raw data of ATP determination for *Figure 3D*.

**Source data 3.** Raw data of quantitative PCR (qPCR) for *Figure 3E*.

mTagBFP-pHluorin was not specifically localized, primarily in the cytosol (*Figure 4A*). After $H_2O_2$ stimulation, a decrease in pHluorin signal was observed, while the signal change was significantly less in Cx43KD cells (*Figure 4B*). ATP5J2, acting as a subunit of ATP synthase, is localized on the mitochondrial inner membrane. We observed most of the ATP5J2-pHluorin signal was merged with MitoTracker Red (*Figure 4—figure supplement 1*). Therefore, the ATP5J2-pHluorin signal indicates the pH changes in the mitochondrial area. After inserting the *ATP5j2* sequence before the NH2-terminal of pHluorin, we monitored the pH signal (GFP/BFP) from *Atp5j2*-mTagBFP-pHluorin transfected MLO-Y4 cells (*Figure 4C*). $H_2O_2$ stimulation evoked a rise in the pHluorin signal in transfected cells (*Figure 4D*). This phenomenon was probably caused by the influx of protons from the mitochondrial intermembrane space to the mitochondrial matrix. The amplitude of the elevation was significantly lower in Cx43KD cells (*Figure 4D*). When plotting signal changes around the mitochondrial area, an oscillation of the mitochondrial pHluorin signal was observed. The amplitude of the oscillation was greatly tapered in Cx43KD mitochondria. Representative plots and the statistical analysis of the maximal amplitude of ATP5J2-pHluorin oscillation are shown in *Figure 4E*. Live-cell imaging results demonstrated that mtCx43 is critical for mitochondrial $H^+$ flux in response to $H_2O_2$ stimuli.

## Inhibition of mtCx43 HCs impedes mitochondrial function

The membrane tropology of Cx43 on the mitochondrial membrane is important, with its C-terminus facing the intermembrane space. For, as we showed Cx43(CT) antibody, not Cx43(E2) antibody, has access to inhibit mtCx43 HCs (*Figure 1F*). For this reason, we used the Cx43(CT) antibody as a specific mtCx43 HC inhibitor in the mitochondrial seahorse coupling assay after the addition of a PMP. Cx43(CT) antibody treatment inhibited the increase of ATP production, similar to CBX, while the rabbit IgG control group did not show a reduction in oxygen consumption (*Figure 5A*). Respiratory control ratios (RCRs, state 3/state 4o) and uncoupling control ratios (UCRs, state 3 u/state 4o) were similar among the groups without any significant difference (*Figure 5B*). The basal respiration (state 2) had no significant difference detected. State 3 after ADP stimulation and state 3u after FCCP uncoupling demonstrated lower OCR in the presence of CT antibody and CBX (*Figure 5C*). We further examined complex I activity using pyruvate and malate as substrates. CT antibody and CBX treated groups showed an inhibitory role compared with Vehicle (Veh) and IgG control groups (*Figure 5D*). CT antibody, and CBX showed lower OCR, indicating reduced complex II activity (*Figure 5E*). The inhibitory role that CT and CBX played was generally observed in Cx43KD MLO-Y4 cells in state 3. ADP-stimulated ATP production significantly decreased in CBX treated group; the CT antibody-treated group displayed a trend of inhibition while no significance was detected (*Figure 5—figure supplement 1*). We also confirmed whether the antibodies did reach mitochondria. After incubating antibodies in the presence of PMP, we stained the cells with a fluorescent anti-rabbit IgG antibody. Cx43(CT) antibody displayed localization on mitochondria (*Figure 5—figure supplement 2*). The signal of CT Ab rather than Cx43(E2) Ab in mitochondria suggested the orientation of mtCx43 CT facing the intermembrane space/cytosolic side, consistent with the mtCx43 orientation reported previously (*Boengler et al., 2009*; *Miro-Casas et al., 2009*).

To further validate the involvement of mtCx43 in proton changes, we generated a Cx43-pHluorin plasmid and expressed it in MLO-Y4 cells (*Figure 6A*). The Cx43-pHluorin signal in live imaging was quantified in the whole cell as well as in the local mitochondrial area after $H_2O_2$ treatment. A decrease in whole-cell Cx43-pHluorin signals calibrated with BFP was observed after adding $H_2O_2$ (*Figure 6B*). In the mitochondrial area, indicated by mitochondrial marker MitoTracker Red, a similar oscillation of pHluorin signal was demonstrated (*Figure 6C*). Since Cx43 was expressed in different subcellular regions, the whole-cell and mitochondrial signals show consistent pHluorin signals in the cytosol and ATP5J2-pHluorin signals in mitochondria. PMP was used to permeate the plasma membrane with intact mitochondria. With a permeable plasma membrane, the Cx43(CT) antibody could reach and function in mitochondria. Our results showed the CT antibody caused a reduction of pHluorin signal oscillation when compared with IgG treatment (*Figure 6D–E*). Although there was a minor effect, the CT antibody did not show significant inhibition of ATP5J2-pHluorin signals (*Figure 6—figure*

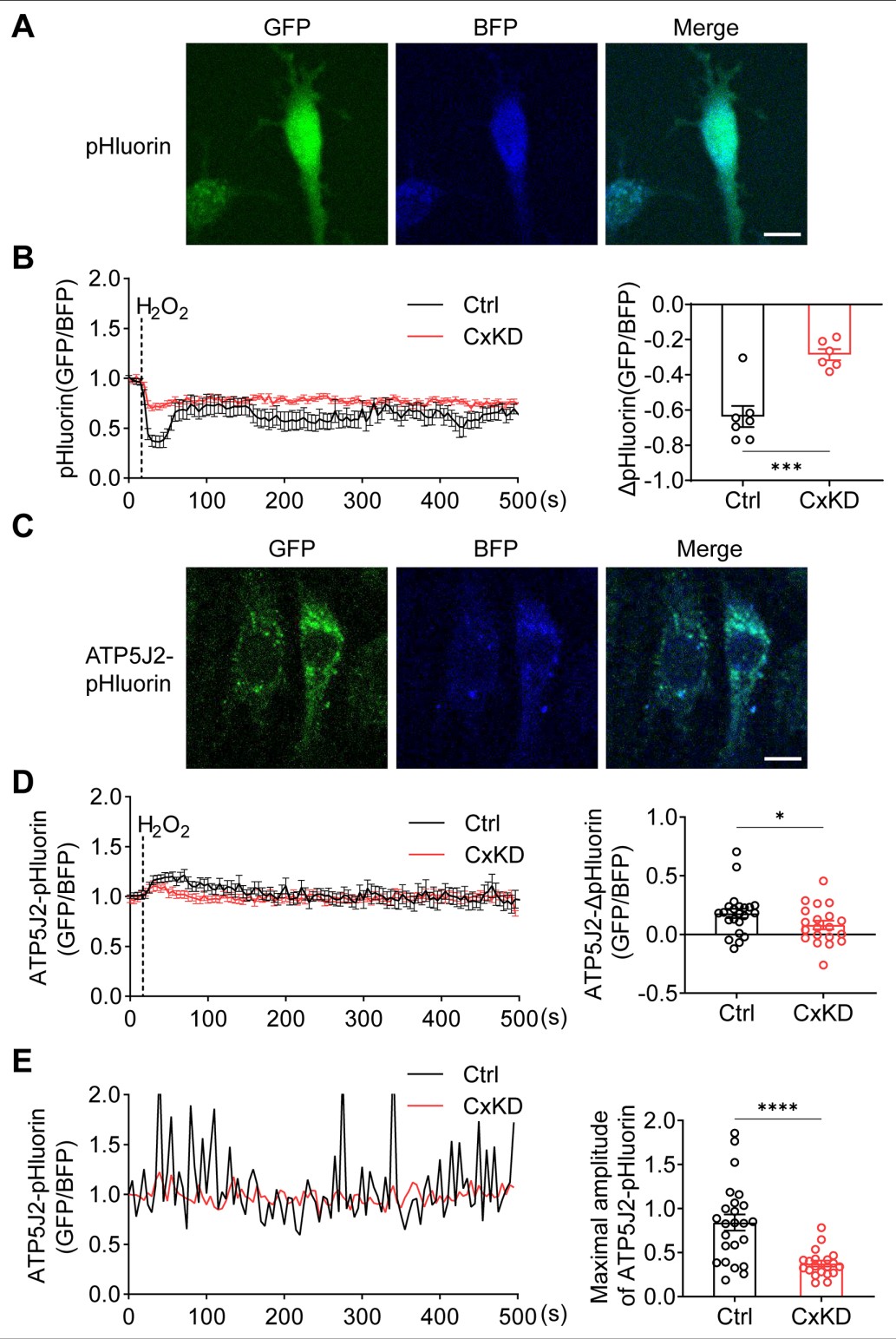

**Figure 4.** pHluorin signal indicated an attenuated sensitivity to oxidative stress in connexin 43 (Cx43) knockdown (KD) Y4 cells. (**A**) pHluorin fluorescent protein localized in the cytoplasmic area of transfected murine long bone osteocyte-Y4 (MLO-Y4) cells. BFP was used as calibrations. (**B**) Cytosolic pHluorin signal change due to $H_2O_2$ stimuli was attenuated in Cx43KD cells. Cytosolic pH change detected in Cx43 KD and control Y4 cells by pHluorin after 0.6 mM $H_2O_2$ stimuli at 25 s time point. Statistical analysis showed the difference in the 40 s time point. ***p<0.001. (**C**) ATP5J2 conjugated pHluorin fluorescent protein localized on mitochondria. Green: pHluorin signal (GFP

*Figure 4 continued on next page*

*Figure 4 continued*

channel); Blue: BFP. Scale bar: 10 μm. (**D**) Mitochondrial pHluorin signal change due to $H_2O_2$ stimuli was attenuated in Cx43KD cells. ATP5J2-pHluorin signal indicated the mitochondrial pH level from Cx43 KD and control Y4 cells after 0.6 mM stimuli at 25 s time point. Statistical analysis showed the difference in the 40 s time point in ATP5J2-pHluorin transfected Y4 cells. Data were collected from n≥7 cells in three independent experiments. *p<0.05. (**E**) Mitochondrial pHluorin signal oscillation was inhibited in Cx43KD Y4 cells. Representative ATP5J2-pHluorin signal oscillation in control and Cx43KD Y4 cells. Two-tailed t-test analysis showed significantly decreased oscillation amplitude in the CxKD group. ****p<0.0001.

The online version of this article includes the following source data and figure supplement(s) for figure 4:

**Source data 1.** Raw data of pHluorin signal in transfected control (Ctrl) and connexin 43 knockdown (CxKD) murine long bone osteocyte-Y4 (MLO-Y4) cells for *Figure 4B*.

**Source data 2.** Raw data of ATP5J2-pHluorin signal in control (Ctrl) and connexin 43 knockdown (CxKD) cells for *Figure 4D&E*.

**Figure supplement 1.** ATP5J2-pHluorin localized on mitochondria.

*supplement 1*). The results showed that the inhibition of mtCx43 HCs by Cx43(CT) antibody caused a decrease in pHluorin signal, indicating a reduced proton gradient across the mitochondrial inner membrane. The increased pH gradient by mtCx43 HCs may be one of the underlying mechanisms that impacts mitochondrial function in response to oxidative stress. Considering that the proton gradient across the mitochondrial inner membrane is essential for ATP synthesis, we investigated the connection between mtCx43 and mitochondrial ATP synthase.

## Cx43 directly interacts with ATP synthase subunit ATP5J2

We showed that mtCx43 HCs are functionally involved in ATP generation. We then determined the spatial relationship between mtCx43 and ATP generation machinery mitochondrial complex V. In complex V, ATP5J2 is the subunit that circles the ATP synthase peripheral stalk with the NH2-terminus facing the matrix and a membrane inserted helix (*Leithe et al., 2018*) and was identified to be associated with Cx43 in human chondrocytes (*Gago-Fuentes et al., 2015*). To investigate if there is physical interaction between the mtCx43 and ATP synthesis protein, we employed the Förster resonance energy transfer (FRET) approach. Cx43-cyan fluorescent protein (CFP) and ATP5J2-enhanced yellow fluorescent protein (EYFP) were co-transfected into MLO-Y4 cells. CFP and YFP form a FRET pair in which CFP works as the donor and YFP as the acceptor (*Miyawaki et al., 1997*). The FRET signal was detected when the two fluorescent proteins were in proximity of less than 10 nm to each other. ATP5J2-EYFP signal was primarily detected in the mitochondria region, while the Cx43-CFP signal was also detected in other parts of the cell. However, the FRET signal was detected (*Figure 7A*), suggesting that Cx43 and ATP5J2 physically interacted with each other in the mitochondria. An immunoprecipitation assay was also conducted using the GFP or ATP5J2-GFP transfected MLO-Y4 cells (*Figure 7—figure supplement 1A*). Cx43 band was detected when an anti-GFP antibody against ATP5J2-GFP was used as an immunoprecipitating antibody (*Figure 7B*). To further investigate the binding between Cx43 and ATP5J2 in mitochondria, we performed a protein pull-down assay with ATP5J2-GFP transfected MLO-Y4 cell extracts using Glutathione s-transferase tag (GST)-Cx43 CT immobilized on glutathione-agarose beads (*Figure 7—figure supplement 1B*). The results showed that GST-Cx43 CT, not GST control, was able to pull down ATP5J2-GFP, further demonstrating a direct interaction of Cx43 CT with ATP5J2 in intermembrane space (*Figure 7C*). Together, these findings suggested a direct binding of complex ATPase subunit ATP5J2 with Cx43 CT in mitochondria.

## Cx43 accumulated in mitochondria of the oxidized osteocytes in *Csf1*± mice in vivo

To assess whether Cx43 migrates and accumulates in mitochondria in response to oxidative stress in vivo, we used a transgenic mouse model with increased oxidative stress in osteocytes. Our previous study suggested that deficiency of macrophage-colony stimulating factor 1 (CSF1), a key factor involved in cell signaling and remodeling, increases Nox4 oxidase expression in primary osteocytes, resulting in elevated oxidative stress (*Werner et al., 2020*). We isolated the primary osteocytes from *Csf1*± and measured the ROS level by DCF staining. We observed a significantly increased ROS production in

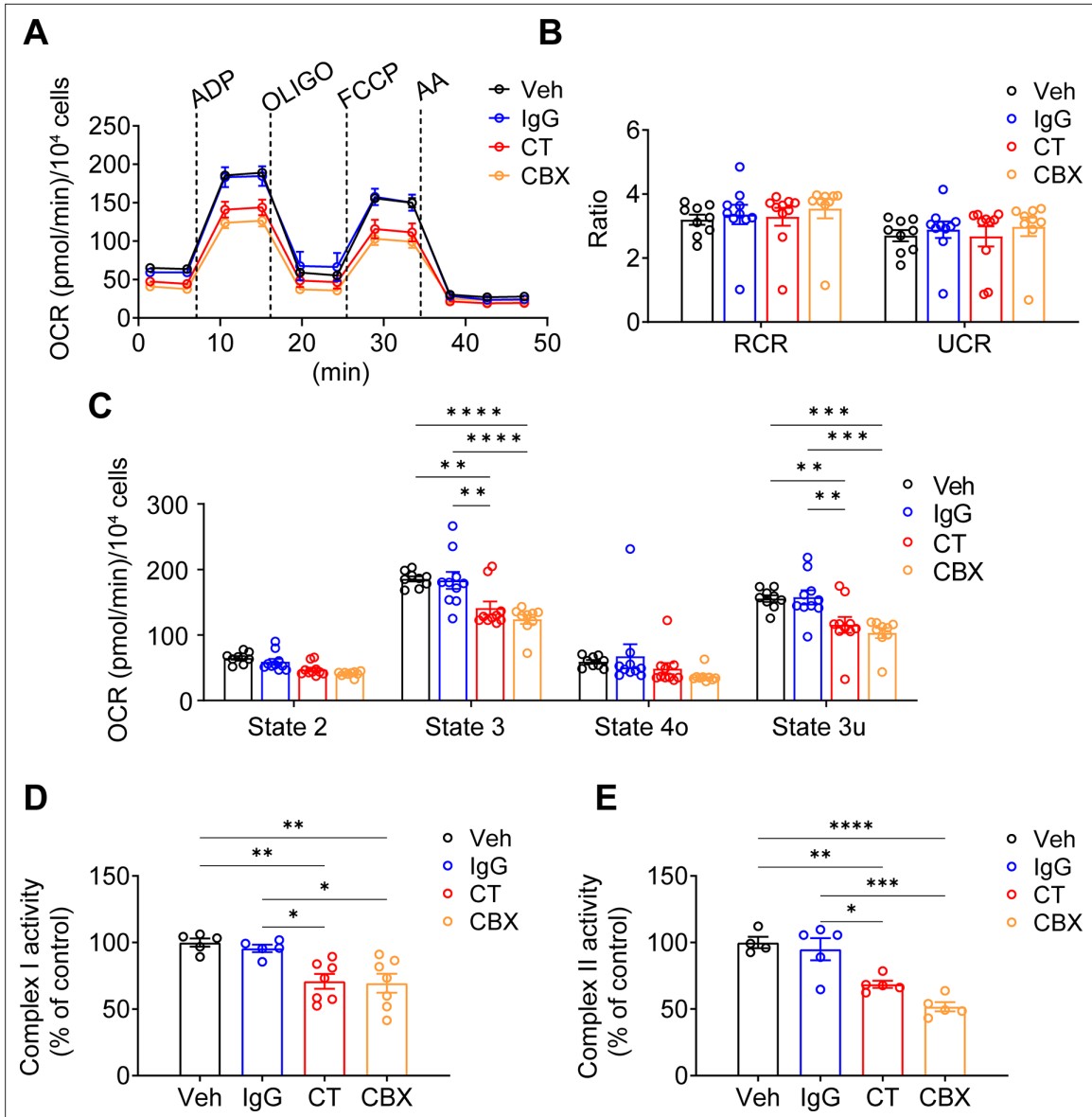

**Figure 5.** Connexin 43 C-terminal (Cx43CT) Ab treatment decreased mitochondrial respiratory capacity in murine long bone osteocyte-Y4 (MLO-Y4) cells. (**A**) Oxygen consumption rate (OCR) was measured on permeabilized MLO-Y4 cells, using an XF96e analyzer. The sequential injection of mitochondrial inhibitors as indicated by arrows. OLIGO, oligomycin; AA, antimycin A. Rabbit IgG, Cx43CT antibody, and carbenoxolone (CBX) were pre-added in the assay medium. Basal respiration (state 2), after the injections of ADP (state 3), OLIGO (state 4o), p-trifluoromethoxy carbonyl cyanide phenylhydrazine (FCCP; state 3u). (**B**) The respiratory control ratio RCR (RCR = state 3/state 4o) and uncoupling control ratio UCR (UCR = state 3u/state 4o), which reflects the mitochondrial respiratory capacity, had no difference between groups. One-way ANOVA test. (**C**) OCR was detected significantly different in state 3 and state 3u. Values corresponding to the different respiratory states are represented as mean ± SEM (n=15–18 replicate of three independent experiments/group). Two-way ANOVA analysis was performed. *p<0.05, ***p<0.001, and ****p<0.001. (**D**) Both C-terminal (CT) antibody and CBX treated groups showed an inhibitory role on complex I activity. Complex I activity was measured in XFe96 Seahorse bioanalyzed by sequential administration of a combination of pyruvate (10 mM)/malate (0.5 mM), FCCP (0.5 µM), and antimycin A (4 µM). The activity was normalized to the control group. *p<0.05, **p<0.01. (**E**). CT antibody and CBX treated group showed a similar inhibitory role on complex II activity. Complex II enzymatic activity was analyzed similarly to Complex I. After basal OCR measurements cells were sequentially treated with a combination of rotenone (2 µM)/succinate (10 mM), FCCP (0.5 µM), and antimycin A (4 µM). All data were reported as the mean ± SEM. *p<0.05, **p<0.01, ***p<0.001, and ****p<0.001.

The online version of this article includes the following source data and figure supplement(s) for figure 5:

**Source data 1.** Raw oxygen consumption rate (OCR) data of Seahorse assay for *Figure 5*.

**Figure supplement 1.** Mitochondrial function was affected under connexin 43 C-terminal (Cx43CT) Ab treatment in Cx43 knockdown (KD) murine long bone osteocyte-Y4 (MLO-Y4) cells.

*Figure 5 continued on next page*

*Figure 5 continued*

**Figure supplement 1—source data 1.** Raw oxygen consumption rate (OCR) data of Seahorse assay for *Figure 5—figure supplement 1*.

**Figure supplement 2.** Connexin 43 (Cx43) C-terminal (CT) antibody localized to mitochondria with plasma membrane permeabilizer (PMP).

primary osteocytes with decreased CSF1 (*Figure 8A*). Furthermore, the significant increase of co-localization of Cx43 and SDHA suggested the increased migration of Cx43 to the mitochondrial in *Csf1*$^{\pm}$ compared with the wild-type (WT) (*Csf1*$^{+/+}$) cells (*Figure 8B*). The result recapitulated the mitochondria migration and accumulation of Cx43 in oxidized osteocytes in vivo.

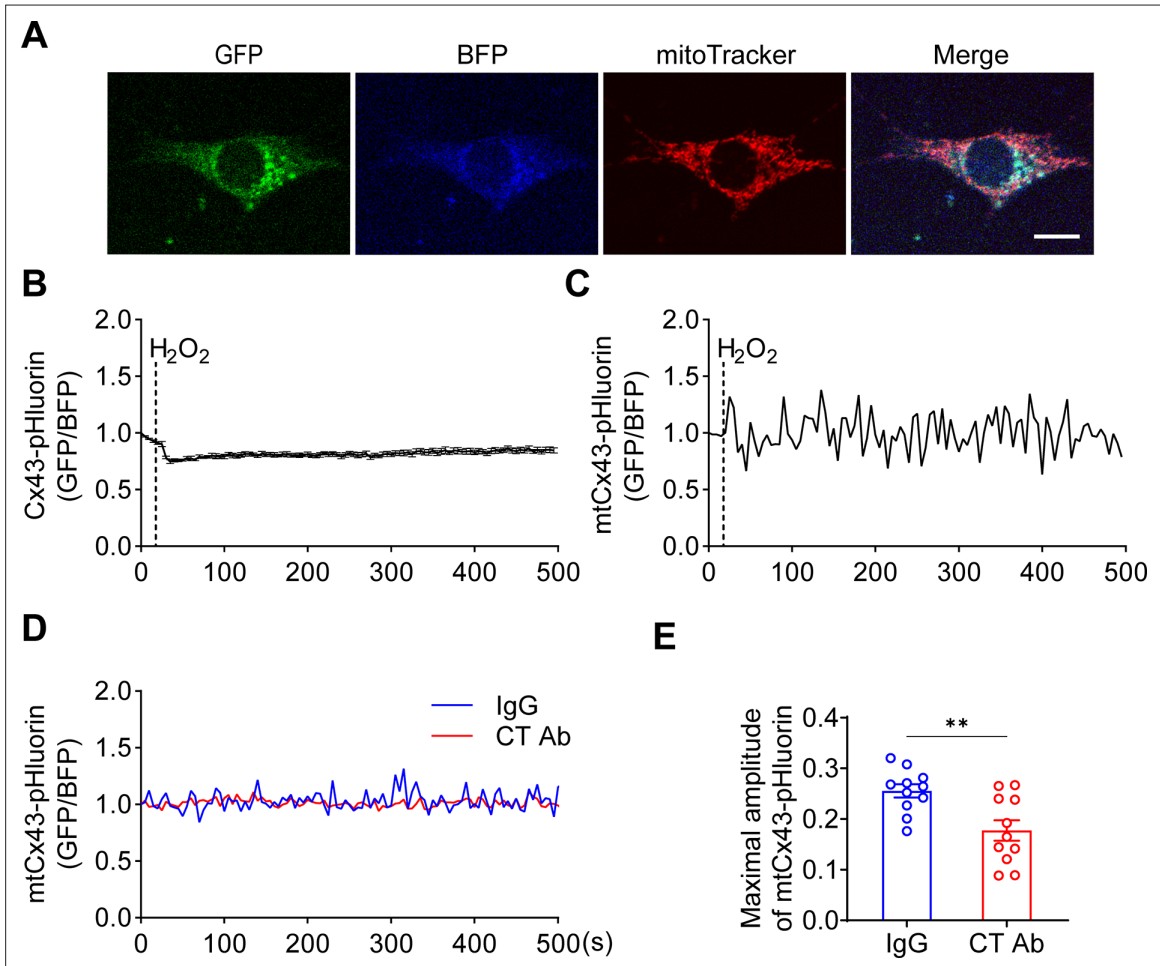

**Figure 6.** Mitochondrial connexin 43 (mtCx43) hemichannels (HCs) were permeable to protons indicated by Cx43-pHluorin signal. (**A**) Typical images of Cx43-pHluorin in murine long bone osteocyte-Y4 (MLO-Y4) cells. Cx43-pHluorin fluorescent protein localization in transfected MLO-Y4 cells. Green: pHluorin signal (GFP channel); Blue: BFP; Red: MitoTracker Red. Scale bar: 10 μm. (**B**) Cx43-pHluorin signal was reduced in whole-cell area after 0.6 mM $H_2O_2$ treatment. BFP signal was used for calibrations. (**C**) Representative mitochondrial Cx43-pHluorin oscillation in MLO-Y4 cells after 0.6 mM $H_2O_2$ treatment. BFP signal was used for calibrations. (**D and E**) C-terminal (CT) antibody reduced Cx43-pHluorin signal oscillation. Mitochondrial pHluorin signal oscillation was recorded in Cx43-pHluorin transfected MLO-Y4 cells. The maximal amplitude of the oscillation was analyzed. **p<0.01.

The online version of this article includes the following source data and figure supplement(s) for figure 6:

**Source data 1.** Raw data of pHluorin signal in the whole cell and the mitochondrial area after connexin 43 (Cx43)-mBFP-pHluorin transfection (*Figure 6B&C*).

**Source data 2.** Raw data of mitochondrial connexin 43 (mtCx43)-pHluorin signal after plasma membrane permeable and treated with IgG and C-terminal (CT) Ab treatment for *Figure 6D&E*.

**Figure supplement 1.** ATP5J2-pHluorin spontaneous oscillation on mitochondria.

**Figure supplement 1—source data 1.** Raw data of ATP5J2-pHluorin signal with IgG or C-terminal (CT Ab treatment for *Figure 6—figure supplement 1*).

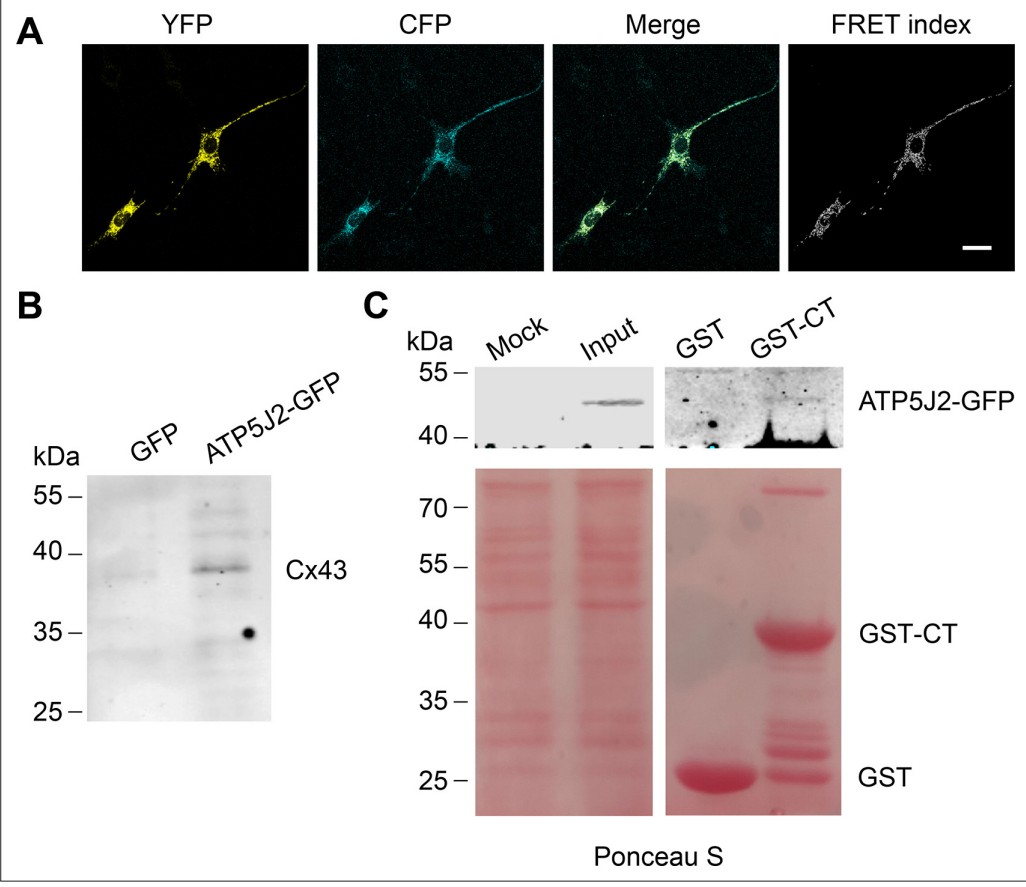

**Figure 7.** Connexin 43 (Cx43) interacted with ATP5J2 on mitochondria. (**A**) Förster resonance energy transfer (FRET) assay showed the interaction between Cx43 and ATP5J2 in murine long bone osteocyte-Y4 (MLO-Y4) cells. MLO-Y4 cells co-expressing Cx43-cyan fluorescent protein (CFP) and ATP5J2-EYFP were presented. YFP fluorescence (EYFP was excited, and its fluorescence recorded; the first column), CFP fluorescence (CFP was excited and its fluorescence recorded; the second column), and 'FRET index' (images processed with the 'FRET and co-localization analyzer,' fourth column). Scale bars: 20 μm. (**B**) Immunoprecipitation showed physically binding of Cx43 with endogenous ATP5J2. Cx43 band was detected after immunoprecipitation using ATP5J2-GFP as the bait. After immunoprecipitation, samples were collected and ran through sodium dodecyl-sulfate polyacrylamide gel electrophoresis (SDS-PAGE). Anti-Cx43 (1:300) was used to incubate the membrane. (**C**) GST pull-down assay proved the physical binding between ATP5J2 and Cx43-CT. Purified GST-CT was conjugated to glutathione agarose beads with GST as control.

The online version of this article includes the following source data and figure supplement(s) for figure 7:

**Source data 1.** Raw WB images of connexin 43 (Cx43) immunoprecipitation (IP) for *Figure 7B*.

**Source data 2.** Raw WB images of GST pull-down assay for *Figure 7C*.

**Figure supplement 1.** Overexpression of endogenous ATP5J2-GFP in murine long bone osteocyte-Y4 (MLO-Y4) cells.

## Discussion

Our study showed that accumulated Cx43 in mitochondria played an important role in protecting mitochondrial and cellular homeostasis against oxidative stress. mtCx43 participates in mitochondrial membrane potential. mtCx43 HCs participate in proton gradient maintenance between the inter-membrane space and matrix, enhancing mitochondrial ATP generation. Moreover, the direct physical interaction between mtCx43 and mitochondrial ATP5J2 demonstrated the coupling of mtCx43 with a subunit of ATP synthase, the mitochondrial complex V. This interaction is likely to have a positive impact on transferring ATP generated by complex V via mtCx43 HCs to increase ATP supply in the cytosol, a critical step for protecting cells against oxidative stress. Together, our data unveiled a new

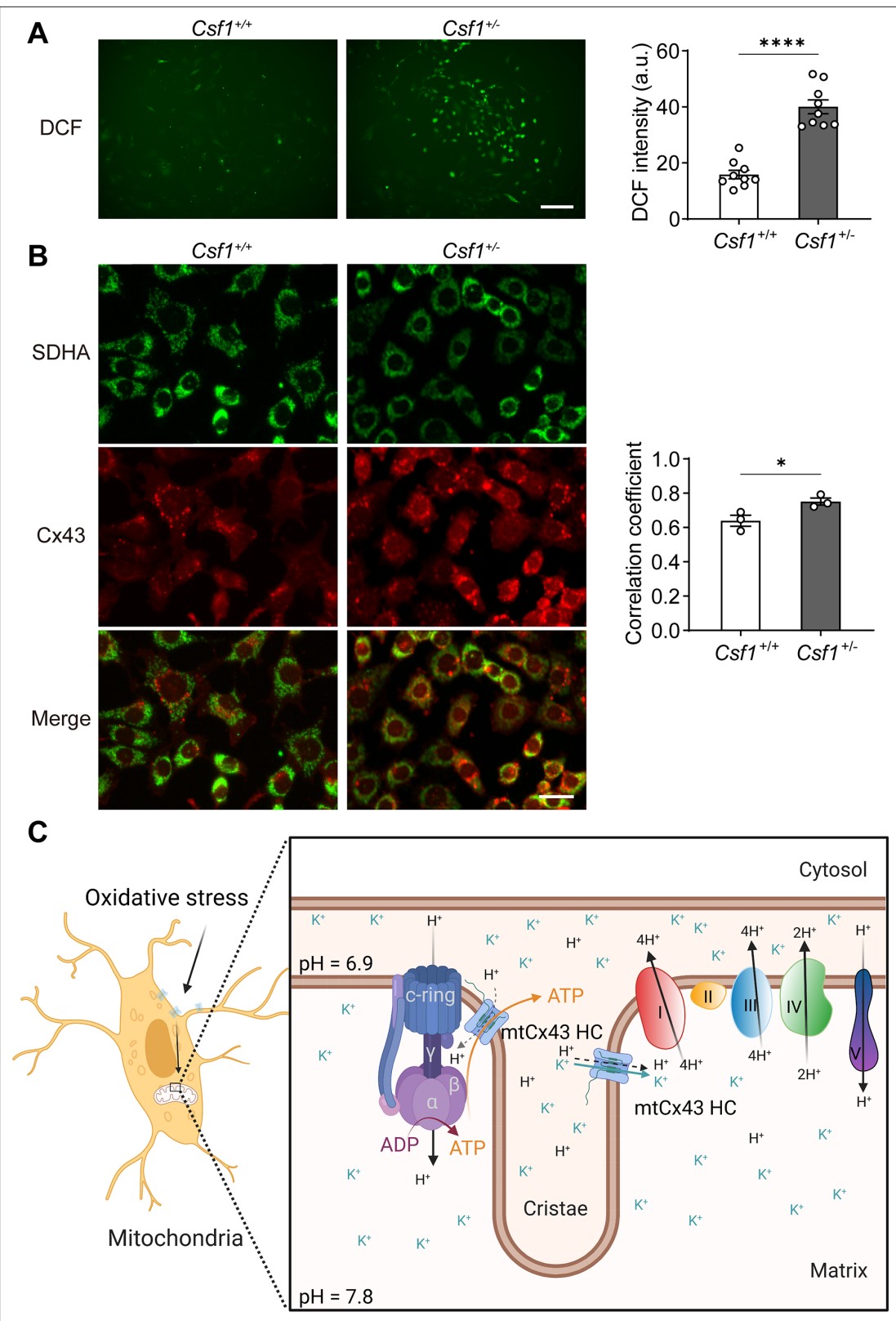

**Figure 8.** Mitochondrial connexin 43 (mtCx43) response to oxidative stress in primary osteocytes. (**A**) Osteocytes from *Csf1*± have an increased reactive oxygen species (ROS) level by DCF staining. Data collected from three animals in each group, each point indicated the ROS level from each individual well. Scale bar: 50 μm. Two-tailed t-test was conducted, *\*\*\*\**p<0.0001. (**B**) Osteocytes from *Csf1*± mice have an increased co-localization of succinate dehydrogenase (SDHA, green) and Cx43 (red). Primary osteocytes were stained with SDHA and Cx43 antibodies. Scale bar: 20 μm. Coefficient co-

*Figure 8 continued on next page*

*Figure 8 continued*

localization analysis of SDHA and Cx43 based on fluorescence signals. Data collected from three mice in each group. Two-tailed t-test, *p<0.05. (**C**) Schematic diagram on mtCx43 function in osteocytes. Organelles (mitochondria and nucleus) are not drawn to scale. In oxidative stress, Cx43 is translocated onto mitochondria. With a special position close to ATP synthase, mtCx43 hemichannels (HCs) enabled the $H^+$ influx between the mitochondrial inner membrane, accelerating ATP production. Thus, mtCx43 HCs play a crucial role in osteocytes under oxidation.

The online version of this article includes the following source data for figure 8:

**Source data 1.** Raw data of DCF staining and succinate dehydrogenase (SDHA), connexin 43 (Cx43) co-localization analysis in primary osteocytes isolated from *Csf1$^{+/+}$* and *Csf1$^{\pm}$* for *Figure 8A and B*.

mechanism concerning the unique role of mtCx43 HCs in mitochondria function, ATP generation, and transport under oxidative insults.

Osteocytes, as the most abundant bone cell types, are dynamic and responsive cells, forming a network in the skeleton to orchestrate bone modeling and remodeling (**Bonewald, 2011**). Oxidative stress is a major insult to bone tissue, and increased oxidative stress during postmenopausal or glucocorticoid medication is one of the major underlying causes of bone loss and osteoporosis (**Domazetovic et al., 2017**). Here, we observed that when an acute oxidant stimulus occurs, Cx43 is rapidly translocated into the mitochondria inner membrane. We used Cx43 KD MLO-Y4 cells to explore the role of mtCx43 and found that mtCx43 expression was tightly correlated with ATP generation. Decreased OCR in permeabilized Cx43 KO MLO-Y4 cells indicated the role of Cx43 in coupling the mitochondrial inner membrane and ATP production. Mitochondria are the fuel factory for cells, producing the bioenergetic fuel ATP for cellular activities (**Spinelli and Haigis, 2018**). Mitochondrial electron transport complexes I–V are responsible for ATP production (**Zhao et al., 2019**). Accompanied by transporting electrons, mitochondrial electronic complexes generate a proton gradient between the mitochondrial intermembrane space and the mitochondrial inner matrix (**Perry et al., 2011**). ATP synthase, also known as complex V, utilizes this proton gradient to produce ATP from ADP to preserve the normal homeostasis and function of cells.

Given the vital role of protons in mitochondria, we utilized pHluorin, as a proton indicator, to investigate the impact of mtCx43 on local proton activity indicated by pH changes. Cx43KD MLO-Y4 cells had reduced pH changes in both cellular and mitochondrial compartments. It has been reported that plasma membrane Cx43 channels permeate protons (**Swietach et al., 2007**). If mtCx43 HCs mediate proton transfer between the intermembrane space and matrix, we would expect to observe an increased proton gradient in Cx43KD cells. Instead, we observed decreased ATP5J2-pHluorin signal and pH gradient in Cx43KD cells. The reduced proton level in the intermembrane space and pH gradient was also detected when mtCx43 HCs were inhibited by Cx43(CT) antibody. Cx43 has four transmembrane domains with a long cytosolic CT. Due to the orientation of mtCx43 HCs with the CT facing the mitochondrial intermembrane space (**Boengler et al., 2009**; **Miro-Casas et al., 2009**), Cx43(CT) specific antibody, rather than Cx43(E2) antibody, functioned as the gating inhibitor for mtCx43 HCs. The addition of PMP, which permeabilizes the plasma membrane, permits the entry of the Cx43(CT) antibody into the cell and targets the mtCx43 in intact mitochondria. The consistent results obtained with Cx43KD and treatment with Cx43(CT) antibody also reduced protons at the intermembrane space and the pH gradient. These results indicate that mtCx43 HCs may not primarily act as leaky proton channels but may instead regulate the ion transport that indirectly increases the proton transport against the proton gradient. Indeed, the mitochondrial membrane potential measured by TMRE was decreased in Cx43KD cells.

In mitochondria, both $H^+$ and $^{.}OH$ gradients across the intermembrane space are essential for the mitochondrial membrane potential (**Lee, 2019**). If taking the small pH chemical gradient into account, the mitochondrial membrane potential is, more correctly, a proton motive force (**Nicholls, 2002**). The electric membrane potential of mitochondria ranges from −150 to 180 mV (**Santo-Domingo and Demaurex, 2012**). Wide-range changes of $H^+$ influx will cause the depolarization of the mitochondrial inner membrane. The opening of mPTP causes a propagated loss of mitochondrial membrane potential. Meanwhile, mPTP opening can be triggered by $Ca^{2+}$ and ROS (**Ong and Gustafsson, 2012**). mtCx43 is reported to engage in the inhibition of mPTP opening, ultimately resulting in the maintenance of ATP production (**Ferko et al., 2019**). Azarashvili et al. also revealed that CBX shortened the lag time before mPTP opening and accelerated the $Ca^{2+}$-induced permeability transition in synaptic and non-synaptic rat brain mitochondria (**Azarashvili et al., 2011**). In adipose tissue, specific KD of

Cx43 leads to a significantly lower expression of components on mitochondrial complexes II, III, and V (*Kim et al., 2017*). The decrease in associated proteins in the ETCs may contribute to the lower energy expenditure.

Cx43 HCs have a large conductance (~220 pS) with relatively low ion selectivity, allowing the passage of ions such as $Ca^{2+}$, $Na^+$, and $K^+$ (*De Smet et al., 2021*; *Wang et al., 2013*). Previous studies indicated the permeability of mtCx43 HCs to $K^+$ (*Boengler et al., 2013*; *Miro-Casas et al., 2009*). The submaximal $K^+$ entry into the mitochondria matrix enhances electron flow with a preserved membrane potential and production of ROS (*Heinen et al., 2007*). To counter the $K^+$ influx, mitochondrial ETC complexes I, III, and IV pump protons out, into the intermembrane space to enhance the proton gradient (*Nolfi-Donegan et al., 2020*). mtCx43 HCs, thus, are likely to mediate the transfer of $K^+$ from the intermembrane space into the matrix. However, we cannot exclude the possibility of the passage of protons following the gradient from the intermembrane space to the matrix. In the intermembrane space, $K^+$ concentration is about 140 mM, while $H^+$ is around 0.1 μM (*Feher, 2012*; *Garlid and Paucek, 2003*). Therefore, the amount of $K^+$ passed through mtCx43 HCs from the intermembrane space to the matrix driven by electrical gradient is expected to be proportionally much greater than that of $H^+$. We did observe the reduction of proton leak in Cx43KD mitochondria, indicating the relatively minor role of mtCx43 HCs in transferring $H^+$ from intermembrane space. pHluorin is a pH sensitive GFP variant, and the use of a fusion protein by linking pHluorin with Cx43 CT permits the detection of proton changes close to Cx43 in the intermembrane space. The local Cx43-pHluorin signal in the mitochondria region displayed an oscillation pattern, suggesting that there is proton flux activity close to mtCx43. Cx43(CT) antibody abolished this oscillation, suggesting mtCx43 HCs regulate proton flux in the intermembrane space. Taken together, the mtCx43 HCs are likely to play a predominant role in transferring $K^+$ into the matrix, thus the proton output into the intermembrane space through electron transport complexes is more significant than proton influx into the matrix, therefore increasing membrane potential, proton gradient, and ultimately generating more ATP.

The majority of the proton leak across the mitochondrial inner membrane is thought to be correlated to the adenine nucleotide translocase and uncoupling proteins (*Divakaruni and Brand, 2011*; *Jastroch et al., 2010*). Proton leak is tightly linked to ROS generation; increased mtROS generated from ETC induces a proton leak, and the proton leak suppresses ROS production as a feedback loop (*Cheng et al., 2017*). There are conflicting studies on proton leak and ROS generation; few reports suggest an increased ROS level after a proton leak (*Boveris and Chance, 1973*; *Starkov and Fiskum, 2003*), while most studies corroborate the positive feedback loop of proton leak and ROS generation (*Brookes, 2005*; *Echtay et al., 2002*). The controversy may be caused by the amplitude of the proton leak, for a mild-moderate proton leak is reported to be protective during I/R injury in cardiomyocytes (*Nadtochiy et al., 2006*). In our case, the proton leak mediated by mtCx43 HCs appears to be at mild-moderated levels and, thus, may play a protective role on osteocytes.

Under normal conditions, few mtCx43 are present in the mitochondria, while under oxidative stress, the accumulation of Cx43 in the mitochondria inner membrane and functional mtCx43 HCs appears to play significant roles in enhancing the ATP generation as well as export of ATP into the cytosol. We showed a significant reduction of ATP in both mitochondria and cytosol with the deletion of Cx43 in osteocytes. Cx43 HCs have been reported to mediate the transport of ATP (*Kang et al., 2008*; *Patel et al., 2014*). Mass spectrometry analysis in mouse hearts indicated the new interaction between Cx43 with apoptosis-inducing factor (AIF) and the beta-subunit of the electron-transfer protein (ETFB). AIF and ETFB are typical proteins regulating mitochondria OXPHOS (*Denuc et al., 2016*). We showed that mtCx43 co-localized and interacted with a subunit of ATPase, ATP5J2 in complex V, and the ATP level in the cytosol was reduced in Cx43KD. It is likely that mtCx43 HCs may enhance the transport of the freshly generated ATP by complex V to the inner membrane space and cytosol. Cx43KD had no effect on the expression of the *Slc25a5* gene, an ATP/ADP translocase, suggesting that increased ATP level in the cytosol is not caused by the impact of mtCx43 on this transporting mechanism. Thereby, the close physical interaction may help accelerate the export of ATP synthesized in the mitochondrial matrix by complex V. The increased ATP level in cytosol will protect the cells against oxidative damage since oxidative stress depletes cytosolic ATP, leading to cell apoptosis. mtCx43's protecting role on a cardiac pre-condition (*Rodríguez-Sinovas et al., 2018*) may also benefit from these mechanisms. At this stage, we cannot exclude other possible roles of mtCx43 that ought to be investigated in the future.

Together, the translocation of Cx43 to mitochondria in osteocytes after oxidative stress acts as a cell 'rescue mission' by promoting ATP production through increased mitochondrial membrane potential and proton gradient through activation of mitochondrial complexes. The localization and interaction of complex V ATPase may aid in the transfer of ATP into the cytosol in protecting cells against oxidative stress (*Figure 8C*).

# Materials and methods

## Key resources table

| Reagent type (species) or resource | Designation | Source or reference | Identifiers | Additional information |
|---|---|---|---|---|
| Strain and strain background (C57BL/6, Male) | *Csf1* knockout (*Csf1⁺/⁻*) mouse; | ***Werner et al., 2020*** | | Laboratory of Dr. Jean Jiang |
| Cell line (*Mus musculus*) | Murine long bone osteocyte-Y4 (MLO-Y4) | ***Kato et al., 1997*** | Kerafast Cat# EKC002, RRID:CVCL_M098 | Cell line maintained in the Laboratory of Dr. Jean Jiang |
| Antibody | Anti-β-actin (mouse monoclonal) | Invitrogen | Cat# MA5-15739, RRID: AB 10979409 | WB 1:5000 |
| Antibody | Anti-connexin 43 C-terminal (Cx43[CT]) (rabbit polyclonal) | ***Siller-Jackson et al., 2008*** | | WB 1:300, IF 1:300 Produced and purified in the Laboratory of Dr. Jean Jiang |
| Antibody | Anti-Cx43(E2) (rabbit polyclonal) | ***Siller-Jackson et al., 2008*** | | WB 1:300, IF 1:300 Produced and purified in Laboratory of Dr. Jean Jiang |
| Antibody | Anti-protein disulphide isomerase (PDI) (rabbit monoclonal) | Cell Signaling Technology (CST) | Cat# 3501 | WB 1:1000 |
| Antibody | Anti-STX6 (rabbit monoclonal) | Cell Signaling Technology (CST) | Cat# 2869 | WB 1:1000 |
| Antibody | Anti-GFP (rabbit polyclonal) | Abcam | Cat# ab290, RRID:AB_303395 | WB 1:2000 |
| Antibody | Anti-SDHA (rabbit polyclonal) | Invitrogen | Cat #459200, RRID:AB_1501830 | WB 1:1000, IF 1:1000 |
| Recombinant DNA reagent | *Atp5j2*-EGFP | This paper | | Transfected 10⁶ cells with 5 µg plasmids |
| Recombinant DNA reagent | *Atp5j2*-EYFP | This paper | | Transfected 10⁶ cells with 5 µg plasmids |
| Recombinant DNA reagent | *Gja1*-EGFP | This paper | | Transfected 10⁶ cells with 5 µg plasmids |
| Recombinant DNA reagent | *Gja1*-CFP | This paper | | Transfected 10⁶ cells with 5 µg plasmids |
| Recombinant DNA reagent | *Gja1*-mTagBFP-pHluorin | This paper | | Transfected 10⁶ cells with 5 µg plasmids |
| Recombinant DNA reagent | *Atp5j2*- mTagBFP-pHluorin | This paper | | Transfected 10⁶ cells with 5 µg plasmids |
| Sequence-based reagent | *Sdha*-F | This paper | PCR primers | GAGATACGCACCTGTTGCCAAG |
| Sequence-based reagent | *Sdha*-R | This paper | PCR primers | GGTAGACGTGATCTTTCTCAGGG |
| Sequence-based reagent | *Atp5j2*-F | This paper | PCR primers | CGAGCTGGATAATGATGCGGGA |

*Continued on next page*

*Continued*

| Reagent type (species) or resource | Designation | Source or reference | Identifiers | Additional information |
|---|---|---|---|---|
| Sequence-based reagent | *Atp5j2*-R | This paper | PCR primers | GCAGTAGCTGAAAACCACGTAGG |
| Sequence-based reagent | *Slc25a5*-F | This paper | PCR primers | ACACGGTTCGCCGTCGTATGAT |
| Sequence-based reagent | *Slc25a5*-R | This paper | PCR primers | AAAGCCTTGCTCCCTTCATCGC |
| Sequence-based reagent | *Gapdh*-F | This paper | PCR primers | CTTCAACAGCAACTCCCACTCTTC |
| Sequence-based reagent | *Gapdh*-R | This paper | PCR primers | TCTTACTCCTTGGAGGCCATGT |
| Peptide, recombinant protein | GST | This paper | | For GST pull-down, 10 μM |
| Peptide, recombinant protein | GST-CT | This paper | | For GST pull-down, 10 μM |
| Commercial assay or kit | Plasma membrane permeabilizer | Agilent | 102504–100 | 1 nM or 2 nM |
| Chemical compound and drug | Lucifer yellow CH dilithium salt | Invitrogen | Cat# L453 | 50 μM |
| Chemical compound and drug | Tetramethylrhodamine-dextran 10,000 MW | Invitrogen | Cat #D1816 | 25 μg/mL |
| Chemical compound and drug | Carboxy H2DCFDA | Invitrogen | Cat #C2938 | 10 μM |
| Chemical compound and drug | MitoTracker Deep Red | Invitrogen | Cat #M33426 | 100 nM |
| Chemical compound and drug | ER-Tracker Blue-White | Invitrogen | Cat #E12353 | 1 μM |
| Chemical compound and drug | Hochest 33342 | Invitrogen | Cat #H21492 | 100 nM |
| Chemical compound and drug | mitoSOX | Invitrogen | Cat #M36008 | 100 nM |
| Software and algorithm | GraphPad Prism 9 | GraphPad | RRID: SCR_000306 | |
| Software and algorithm | ImageJ | ImageJ | RRID: SCR_003070 | |

## Cell culture and reagents

MLO-Y4 osteocyte cells established from primary osteocytes were cultured on type- I rat tail collagen-coated dishes in α-modified essential medium with 2.5% fetal bovine serum (FBS) and 2.5% bovine calf serum (BCS) with penicillin-streptomycin (PS; *Kato et al., 1997*). The authenticity of the MLO-Y4 cell line was confirmed by examining cell morphology with dendritic processes and maker expression, including high amount of osteocalcin, Cx43 and CD44, and low amount of alkaline phosphates. No mycoplasma was detected. The KD of *Gja1* by CRISPR-Cas9 system in MLO-Y4 cells with *Rosa26* KD as a control was done as previously described (*Hua et al., 2022*). Transfected MLO-Y4 cells were cultured without PS. Cx43(E2) antibody and Cx43(CT) antibody were both developed in the Jiang laboratory (*He et al., 1999*; *Siller-Jackson et al., 2008*). Cx43(E2) antibody has been used as a Cx43 HC inhibitor as previously described (*Kar et al., 2013*; *Sáez et al., 2013*; *Siller-Jackson et al., 2008*). Mouse anti-SDHA antibody was purchased from Invitrogen (#459200, Invitrogen, Waltham, MA, USA).

## Immunofluorescence

Cells were seeded in cell culture plates or 12 mm glass coverslips for 2 days. Cells were then stained in 37°C for 20 min using MitoTracker Deep Red (200 nM, Invitrogen, Waltham, MA, USA), if needed, before fixation with 4% paraformaldehyde for 10 min at room temperature (RT). After washing in PBS buffer for three times, cells were incubated with affinity-purified antibodies against Cx43(E2)

(1:300 dilution) or SDHA (1:1000 dilution) in blocking buffer (0.25% Triton X-100, 2% gelatin, 2% donkey serum, and 1% bovine serum albumin) for 1 hr at RT, followed by incubation with fluorescein-conjugated secondary antibody for 1 hr at RT. After incubation with 4, 6-diamidino-2-phenylindole for 5 min, the cells were rinsed with PBST (phosphate-buffered saline with 0.1% Tween-20) for three times. After mounting, the slides were imaged using a fluorescence microscope (Keyence BZ-X710, Tokyo, Japan) or a confocal microscope (Carl Zeiss, Thornwood, NY, USA).

## Protein extraction and immunoblot analysis

Membrane protein extracts from MLO-Y4 cells were prepared for western blot and GST pull-down assays. Cells were lysed and homogenized by a 20 G needle with ice-cold lysis buffer (5 mM EDTA, 5 mM EGTA, and 5 mM Tris, pH 8.0). Proteinase inhibitors (2 mM phenylmethylsulfonyl fluoride (PMSF), 5 mM N-ethylmaleimide (NEM), 1 mM $Na_3VO_4$, and 0.2 mM leupeptin) were added. The cell lysates were then centrifuged at 600×g for 5 min, removing intact cells and nuclei. The supernatant was then centrifugated (Beckman Coulter, Brea, CA, USA) at 125,000×g for 30 min at 4°C. The pellets were then resuspended in lysis buffer with 1% SDS for western blot or in 0.5% Triton X-100 buffer for GST pull-down assay.

Protein concentration was quantified using the micro-BCA assay (Pierce, Rockford, IL, USA), and equal amounts of proteins were loaded on a SDS-polyacrylamide gel electrophoresis (PAGE) and electroblotted onto a nitrocellulose membrane. Membranes were incubated first with primary antibodies, anti-Cx43(CT) (1:300 dilution), anti-SDHA (1:1000, Invitrogen), and anti-β actin antibodies (1:5000) overnight at 4°C, followed with the incubation with secondary antibodies, goat anti-rabbit IgG conjugated IRDye 800CW, or goat anti-mouse IgG conjugated IRDye 680RD (1:15,000 dilution), 1 hr at RT. The images were captured using a Licor Odyssey Infrared Imager (Lincoln, NE, USA), and protein band intensity was quantified using NIH Image J software.

## DNA construct preparation and transfection

The coding region of mouse Cx43 (*Gja1*) was amplified by PCR, and DNA fragments were inserted into the EGFP-N1 or CFP-N1 vector to generate Cx43 (*Gja1*)-EGFP or Cx43 (*Gja1*)-CFP, respectively. *Atp5j2* cDNA was inserted into the EGFP-N1and EYFP-N1 vector to generate *Atp5j2*-EGFP or *Atp5j2*-EYFP, respectively. The mTagBFP-pHluorin plasmid was used to generate *Cx43*-mTagBFP-pHluorin and *Atp5j2*-mTagBFP-pHluorin constructs. For live-cell imaging, MLO-Y4 cells were transfected with *Gja1*-mTagBFP-pHluorin or *Atp5j2*-mTagBFP-pHluorin using the Neon transfection system with parameters set as 1200 mV with a duration of 20 ms, 1 pulse. The ratio of cells and transfected plasmids was $10^6$ cells with 5 μg plasmids. The expression of endogenous plasmids was determined by fluorescence imaging intensity.

## Mitochondria isolation

MLO-Y4 cells were collected, washed with ice-cold PBS twice, and centrifuged for at 1000×g for 5 min at 4°C. The pellets were resuspended in 1 mL of mannitol buffer (220 mM mannitol, 70 mM sucrose, 20 mM HEPES, 1 mM EGTA, 0.1% BSA, and adjusted to pH 7.2 with KOH) and then homogenized. The homogenate was centrifuged at 1000×g for 10 min to remove intact cells and nuclei, and the supernatant was then centrifuged at 9000×g for 15 min to precipitate down mitochondria. The pellet containing mitochondria was washed once with mannitol buffer and was resuspended in mannitol buffer with 1% BSA for further usage.

Mitochondrial purity validation. Isolated mitochondria extracts were loaded as well as the whole cell lysates on 12% SDS-PAGE and immunoblotted with SDHA antibody (1:1000, Invitrogen), protein disulphide isomerase antibody (1:1000, #3501, CST, Boston, MA, USA), syntaxin 6 (STX6) antibody (1:1000, #2869, CST, Boston, MA,USA) at 4°C overnight. Isolated mitochondria were incubated with fluorescent dyes specific to different organelles; MitoTracker Deepred (1 μM, # M22426, Invitrogen, Waltham, MA, USA), ER-Tracker Blue-White (1 μM, # E12353, Invitrogen, Waltham, MA, USA) and Hochest 33,342 (100 nM, # H21492, Invitrogen, Waltham, MA, USA) for 30 min at 37°C. Isolated mitochondria were resuspended in 1× isotonic mitochondrial assay solution (220 mM mannitol, 70 mM sucrose, 10 mM $KH_2PO_4$, 2 mM HEPES, 5 mM $MgCl_2$, 1 mM EGTA, 0.2% BSA, and adjusted pH 7.2 with KOH), and after rinsing two times, confocal images were captured under a 63× objective lens.

## mtROS measurement

Intracellular ROS was determined using mitoSOX (#M36008, Invitrogen, Waltham, MA, USA) in intact MLO-Y4 cells or using fluorescence-based probes Carboxy-$H_2$DCFDA (#C2938, Invitrogen, Waltham, MA, USA) in isolated mitochondria. Cells were rinsed with Hanks' balanced salt solution (HBSS) three times and incubated with 100 nM mitoSOX for 20 min at 37°C. After three washes with HBSS, fluorescence images were immediately captured using a Keyence BZ-X710 microscope. Isolated mitochondria were stained with 10 µM Carboxy-H2DCFDA for 10 min at 37°C, and the data were collected using a microplate reader (Bio-Tek Synergy HT, Winooski, VT, USA).

## Mitochondrial dye uptake assay

Mitochondria dye uptake assay was performed based on published studies (*Miro-Casas et al., 2009*; *Soetkamp et al., 2014*). Isolated mitochondria from MLO-Y4 cells were resuspended in isosmotic succinate buffer (150 mM KCl, 7 mM NaCl, 2 mM $KH_2PO_4$, 1 mM $MgCl_2$, 6 mM MOPS, 6 mM succinate, 0.25 mM ADP, 0.5 µM Rot, and 1 nM PMP with pH 7.2) at 0.4 mg/mL. Mitochondria were then aliquoted into five groups, with the addition of either vehicles or Cx43 HC blockers. After a 20 min incubation period at RT and 200×g centrifugation for 3 min, 50 µM of the Cx43 HC-permeable dye LY CH dilithium salt (LY, #L453, Invitrogen, Waltham, MA, USA) and 25 µg/mL of the HC impermeable dye Tetramethylrhodamine-dextran 10,000 MW (TRITC-dextran, #D1816, Invitrogen, Waltham, MA, USA) were added. Samples were incubated for 25 min and centrifuged at 200×g at RT. Mitochondria were subsequently washed for two times and gently resuspended in 200 µL succinate buffer. Fluorescence of LY ($\lambda_{ex}$ 430 nm/ $\lambda_{em}$ 535 nm) and TRITC- dextran ($\lambda_{ex}$ 545 nm/ $\lambda_{em}$ 600 nm) was measured by a microplate reader (Bio-Tek Synergy HT, Winooski, VT).

## Mitochondrial function assay

Mitochondrial OCRs were measured using a Seahorse XFe96 analyzer (Seahorse Bioscience, North Billerica, MA, USA), which was equilibrated at 37°C overnight. $5×10^3$ MLO-Y4 cells were seeded in a pre-coated XF96 plate and cultured overnight for adherence. On the same day of the experiment, 50 µL containing 1× mitochondrial assay solution (220 mM mannitol, 70 mM sucrose, 10 mM $KH_2PO_4$, 2 mM HEPES, 5 mM $MgCl_2$, 1 mM EGTA, 0.2% BSA, and adjusted pH 7.2 with KOH) with 10 mM succinate and 2 µM Rot as substrates for the coupling assay was added to each well of the XF96 plate and followed by addition of PMP (Invitrogen) with a final concentration 2 nM. To assess various parameters of mitochondrial function, OCRs were measured after injecting ADP, OLIGO (a complex V inhibitor), FCCP (a protonophore and mitochondrial uncoupler), and AA (a complex III inhibitor). The final concentrations of drugs in the well were: ADP, 5 mM; OLIGO, 5 µM; FCCP, 5 µM; AA, 10 µM; and Rot, 2 µM. The OCR was measured using the Seahorse XFe96 extracellular flux analyzer (*Andersen et al., 2019*; Au - *Traba et al., 2016*).

## pHluorin live-cell imaging

Transfected MLO-Y4 - pHluorin plasmid cells were used for live-cell imaging. Cells were maintained in a recording medium, and images were captured with a Zeiss LSM-810 confocal laser scanning microscope with ZEN imaging software. Transfected cells with good pHluorin signal were first located under a 10× objective, and the images were then acquired using a 40× water objective. Images were acquired at a frame size of 512×512 pixels with a time interval of 5 s between each frame.

## Protein pull-down assay

GST fusion protein containing Cx43 CT and GST control without fusion protein was individually expressed in *Escherichia coli* and purified by binding to glutathione-coupled agarose beads as described in published procedures (*Jiang et al., 1994*). By using purified GST and recombinant GST-CT fusion protein, we performed the GST pull-down assay (*Sambrook and Russell, 2006*). MLO-Y4 cells transfected with *Atp5j2*-EGFP were cultured in a 10 cm culture plate, and cells were collected after 2 days of culturing. Crude cell membrane was prepared and resuspended in a buffer assay medium (20 mM Tris, 50 mM NaCl, and pH 7.5) containing nonionic (NP-40, Triton X-100) and zwitterionic (CHAPS). The crude membrane extracts were used in the pull-down experiments. The glutathione agarose beads (Piece, IL) were washed, equilibrated with PBS, and aliquoted to 50 µL of the beads into two microcentrifuge tubes. The beads were incubated with 20 µg GST-CT or the same amount

of GST control for 1 hr at 4°C. An equal amount (1 mg) of crude membrane extracts was then added to the tubes containing GST-CT and the GST vials. The mixture was incubated overnight at 4°C. The glutathione agarose beads were washed with PBS five times for 5 min each. An elution buffer (20 mM Tris, pH 8.0) containing 10 mM reduced glutathione was used to elute the proteins bound to the beads. Together with the input samples, the eluted fractions were analyzed using 10% SDS–PAGE, and immunoblotting was done using an anti-GFP antibody (1:2000, #ab290, Abcam, Cambridge, MA, USA).

## FRET microscopy

For FRET microscopy, MLO-Y4 cells were transfected with ATP5J2-YFP and Cx43-CFP plasmids. After 48 hr of culturing, cells were imaged with an LSM-810 confocal microscope and an argon laser (Carl Zeiss, Thornwood, NY, USA) at excitation wavelength 458 nm for CFP and 514 nm for YFP, and emission wavelength 470–500 nm for CFP and 530–600 nm for YFP and FRET channels. Confocal images were acquired sequentially in CFP and YFP channels. Identical parameters were used within each channel as well in all test groups. Images were processed using the FRET analysis plugin in NIH ImageJ software.

## Isolation of primary osteocytes from long bone

Primary osteocytes from $Csf1^{+/+}$ (WT) to $Csf1^{±}$ were prepared as previously described (*Werner et al., 2020*). Briefly, femurs and tibias were isolated from 21-day-old male mice and bone marrow removed. The soft tissue attached on the bone surface was also carefully removed. The cortical bone was cut into 1 mm pieces and then incubated with 300 U/mL collagenase type 1 (Sigma, St. Louis, MO, USA) solution for three times, 5 min each, and digests were discarded. Then, bone pieces were incubated with either 4 mM EDTA or collagenase at 37°C for 30 min twice to remove other cell types. The remaining cortical bone was then incubated with 4 mM EDTA and then with collagenase, collecting the fractions after fraction 7. These later digests were filtered through a 100 µm cell strainer and were enriched osteocytes. The primary osteocytes were cultured on coated dishes with α-MEM (#12561056, Life Technologies, Carlsbad, CA, USA) supplemented with 5% FBS and 5% BCS (Hyclone, Logan, UT, USA). All animal protocols were performed following the National Institutes of Health guidelines for care and use of laboratory animals and approved by the UT Health San Antonio Institutional Animal Care and Use Committee.

## RNA extraction and RT-qPCR

Total RNA was isolated from MLO-Y4 cells using TRIzol reagent (#15596026, Invitrogen, Waltham, MA, USA). cDNA libraby was established by a high-capacity cDNA reverse transcription kit (#4388950, Applied Biosystems, Carlsbad, CA, USA) according to the manufacturer's instructions. mRNA level was analyzed with a two-step amplification (94°C for 10 s and 60°C for 30 s, 40 cycles) using an ABI 7900 PCR device (Applied Biosystems, Bedford, MA, USA). The primers for RT-qPCR are: *Sdha*, Forward 5′-GAGATACGCACCTGTTGCCAAG-3′, Reverse 5′-GGTAGACGTGATCTTTCTCAGGG-3′; *Atp5j2*, Forward 5′-CGAGCT GGATAATGATGCGGGA-3′, Reverse 5′-GCAGTAGCTGAAAACCACGT AGG-3′; *Slc25a5*, Forward 5′-ACACGGTTCGCCGTCGTATGAT-3′, Reverse 5′-AAAGCC TTGCTCCCT TCATCGC-3′; *Gapdh*, Forward 5′-CTTCAACAGCAACTCCCAC TCTTC-3′, Reverse 5′-TCTTACTC CTTGGAGGCCATGT-3′.

## Statistical analysis

Statistical analysis was conducted using GraphPad Prism 9 software (GraphPad Software, La Jolla, CA, USA). Two-tailed t-test was conducted to the comparison between two groups. Multiple group comparisons were done using one-way ANOVA. Two-way ANOVA and Tukey multiple comparisons were used to compare the mean differences between two independent variable groups. The data were presented as mean ± SEM of at least three measurements. $p < 0.05$ was designated as a statistically significant difference. *$p < 0.05$; **$p < 0.01$; ***$p < 0.001$; ****$p < 0.0001$. Images were analyzed using NIH Image J software.

## Materials availability

The newly created plasmids generated in this study are available from the lead contact with a completed materials transfer agreement without restriction.All data were included in the manuscript. Source data has been provided.

## Acknowledgements

We thank Professor Yulong Li at Peking University for generously providing the pHluorin plasmid and Dr. Lynda Bonewald at Indiana University for generously providing MLO-Y4 cell line. This work was supported by the National Institutes of Health (NIH) Grants: CA148724 (to FMA), and 5RO1 AR072020 (to JXJ), and Welch Foundation grant: AQ-1507 (to JXJ).

## Additional information

### Competing interests

Jean X Jiang: Reviewing editor, *eLife*. The other authors declare that no competing interests exist.

### Funding

| Funder | Grant reference number | Author |
| --- | --- | --- |
| National Institutes of Health | CA148724 | Francisca M Acosta |
| National Institutes of Health | RO1 AR072020 | Jean X Jiang |
| Welch Foundation | AQ-1507 | Jean X Jiang |

The funders had no role in study design, data collection and interpretation, or the decision to submit the work for publication.

### Author contributions

Jingruo Zhang, Conceptualization, Data curation, Formal analysis, Investigation, Methodology, Writing - original draft; Manuel A Riquelme, Conceptualization, Formal analysis, Methodology, Writing – review and editing; Rui Hua, Data curation, Methodology, Writing – review and editing; Francisca M Acosta, Validation, Methodology, Writing – review and editing; Sumin Gu, Conceptualization, Methodology, Writing – review and editing; Jean X Jiang, Conceptualization, Supervision, Funding acquisition, Writing - original draft, Project administration, Writing – review and editing

### Author ORCIDs

Jingruo Zhang http://orcid.org/0000-0002-3786-9856
Manuel A Riquelme http://orcid.org/0000-0002-1915-0434
Francisca M Acosta http://orcid.org/0000-0002-0171-9901
Jean X Jiang http://orcid.org/0000-0002-2185-5716

### Decision letter and Author response

Decision letter https://doi.org/10.7554/eLife.82206.sa1
Author response https://doi.org/10.7554/eLife.82206.sa2

## Additional files

### Supplementary files

• MDAR checklist

### Data availability

All data generated or analysed during this study are included in the manuscript and supporting file; Source Data files have been provided.

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
