## [Editor Report]

This fundamental work advances our understanding of the functional role of connexin43 in mitochondrial. The evidence support that it forms hemichannels at the mitochondrial inner membrane, and serves for optimal mitochondrial metabolism, which is enhanced by oxidant stress. The authors expanded their observation to another cell type and provided solid method explanations and experimental evidence on the quality of their mitochondria preparation.

---

## [Decision Letter]

**Decision letter after peer review:**

Thank you for submitting your article "Connexin 43 Hemichannels Regulate Mitochondrial ATP Generation, Mobilization, and Mitochondrial Homeostasis against Oxidative Stress" for consideration by *eLife*. Your article has been reviewed by 3 peer reviewers, including Juan Carlos Saez as Reviewing Editor and Reviewer #1, and the evaluation has been overseen by Mone Zaidi as the Senior Editor. The following individual involved in review of your submission has agreed to reveal their identity: Matthias M Falk (Reviewer #3).

Essential revisions:

1) Please provide evidence of the purity of mitochondrial preparation.

2) Please explain how the anti-Cx43 reaches the C-termini in the mitochondrial intermembrane space. The manuscript would also benefit from additional detail related to the PMP protocol.

3) Please provide figure 1 with better resolution.

4) If experimental evidence is missing, please avoid overstatements. For example, in the abstract, the last statement (…, leading to enhancing the protection capacity of osteocytes against oxidative insults).

5) The manuscript requires general editing.

For example, Page 5. Line 10. The significant increase observed at 2 h is described twice in the same sentence. Line 12. LY is permeant Cx43 HCs or Cx43 HCs are permeable to LY. "Connexins (Cxs) family consists of 20 members". "We found that in mitochondria isolated after H2O2 treatment, the dye uptake increment was inhibited by CBX". "Proton gradient between intermembrane space and matrix is a key process for ATP generation". "Our study showed that accumulated Cx43 in mitochondria played an important role in protecting mitochondria and cell homeostasis agonist oxidative stress." "Under normal conditions, very fewer mtCx43 are present in the mitochondria,"

In Figure 8 "Cristea" should be "Crista" (or "Cristae", if plural).

In Figure 2E, mention that TMRE fluorescence is shown in red.

– What does the X-axis indicate in Figures 3A, 6B, and 6C?

– Heading P7: Add "electron transport chain" to the heading to prevent confusion with GJ coupling.

– Figure 8C: Mention that Organelles (mitochondria and nucleus) are not drawn to scale.

– p. 4, last line: no "how".

– p. 8, line 15: "marker" proteins, not marked.

– p. 13, line 15: "mitochondria", not mitochondrial.

– p. 15, line 6: plasma "membrane" Cx43 channels.

– Sections of M and M need proof-reading.

---

## [Author Response]

Essential revisions:1) Please provide evidence of the purity of mitochondrial preparation.

Additional experiments determining the purity of mitochondrial preparation were performed and the results are shown in a new figure (Figure 1—figure supplement 1). Specifically, we performed western blot to identify the components within the isolated mitochondria fraction and used protein disulfide isomerase (PDI) as the marker of the endoplasmic reticulum (ER), and SDHA for mitochondria, and syntaxin 6 (STX6) for Golgi. The result showed some contamination of ER and Golgi in mitochondrial fractions. Considering the clear enrichment in the mitochondrial fraction indicated by SDHA, we then performed direct staining in isolated mitochondria using fluorescent dyes for different organelles and the amount of fluorescence was examined by confocal imaging. The result indicated that the isolated mitochondrial component was mostly pure with minimal ER contamination (Figure 1—figure supplement 1BandC).

2) Please explain how the anti-Cx43 reaches the C-termini in the mitochondrial intermembrane space. The manuscript would also benefit from additional detail related to the PMP protocol.

We used PMP as a plasma membrane permeabilizer, which permeabilizes the plasma membrane. Meanwhile, PMP has a 6x higher affinity efficiency with the mitochondrial outer membrane (MOM) compared to the mitochondrial inner membrane (MIM), according to the manufacturer. In addition, MOM has much higher permeability than MIM. It is likely that the antibody can reach the intermembrane space. To prove this, we conducted new experiments using PMP-treated MLO-Y4 cells and treated cells with IgG, CT and/or E2 antibodies. After staining with Alexa 488 conjugated anti-Rabbit IgG secondary antibody, we directly observed the signal under a confocal microscope (Figure 5—figure supplement 2A). Cx43 CT signal was overlapped with the MitoTracker signal (Figure 5—figure supplement 2B), IgG group did not have any signal, and Cx43(E2) treated group have some signals, but they are not co-localized in mitochondria. From this observation, we proved the localization of the Cx43(CT) antibody at mitochondria as well as the orientation of mtCx43 with CT facing the intermembrane space/cytosolic side. We clarified the detailed concentration of PMP used in the MandM section. We observed that membrane was permeabilized soon after PMP treatment.

3) Please provide figure 1 with better resolution.

We have replaced the previous images with new confocal images of higher magnification and resolution.

4) If experimental evidence is missing, please avoid overstatements. For example, in the abstract, the last statement (…, leading to enhancing the protection capacity of osteocytes against oxidative insults).

Thank you for the suggestion; we have corrected this overstatement in our manuscript (…, which helped protect osteocytes against oxidative insults) Page 2, line 19, alongside a though overview to ensure the accuracy of statements throughout.

5) The manuscript requires general editing.For example, Page 5. Line 10. The significant increase observed at 2 h is described twice in the same sentence. Line 12. LY is permeant Cx43 HCs or Cx43 HCs are permeable to LY. "Connexins (Cxs) family consists of 20 members". "We found that in mitochondria isolated after H2O2 treatment, the dye uptake increment was inhibited by CBX". "Proton gradient between intermembrane space and matrix is a key process for ATP generation". "Our study showed that accumulated Cx43 in mitochondria played an important role in protecting mitochondria and cell homeostasis agonist oxidative stress." "Under normal conditions, very fewer mtCx43 are present in the mitochondria,"

We have done an extensive overview of the manuscript correcting any mistakes in phrasing, flow, and general edits, alongside having specifically modified the mentioned sentences accordingly.

In Figure 8 "Cristea" should be "Crista" (or "Cristae", if plural).

Thank you for catching this error. We have corrected it in the updated Figure 8.

In Figure 2E, mention that TMRE fluorescence is shown in red.

We have mentioned the color of TMRE in the figure legend alongside adding a label within the figure to enhance clarification.

– What does the X-axis indicate in Figures 3A, 6B, and 6C?

The X-axis in Figure 3A is minutes (min); In 6B and 6C is seconds (s). We have clarified the X-axis in the corresponding figures and updated them.

– Heading P7: Add "electron transport chain" to the heading to prevent confusion with GJ coupling.

We have modified the title to: “Mitochondrial electron transport chain coupling is impaired after Cx43 KD.”

– Figure 8C: Mention that Organelles (mitochondria and nucleus) are not drawn to scale.– p. 4, last line: no "how".– p. 8, line 15: "marker" proteins, not marked.– p. 13, line 15: "mitochondria", not mitochondrial.– p. 15, line 6: plasma "membrane" Cx43 channels.– Sections of MandM need proof-reading.

We have corrected the writing accordingly and extensively proofed the Materials and methods sections.